# Brown adipose tissue activity impacts systemic lactate clearance in male mice

Rémi Montané[1], Yannick Jeanson[1] (iD), Damien Lagarde[1], Spiro Khoury[1,2] (iD), Léana Porcher-Bibes[1,2],
Jean Nakhle[1], Marie Sallese[1], Mélissa Parny[1,3] (iD), Isabelle Raymond-Letron[1,3], Emma Huard[1] (iD),
Raphael Alves de Souza[1,4], Anne Galinier[1], Luc Pellerin[5] (iD), Anne-Karine Bouzier Sore[6],
Jean-Philippe Pradère[1], Cédric Moro[7] (iD), Louis Casteilla[1], Armelle Yart[1] (iD), Cédric Dray[1,8] (iD),
Jean-Charles Portais[1,2,9], Isabelle Ader[1,8] (iD) and Audrey Carriere[1] (iD)

[1]*RESTORE Research Center, INSERM 1301, CNRS 5070, EFS, ENVT, Université de Toulouse, Toulouse, France*
[2]*MetaboHUB-MetaToul, National Infrastructure of Metabolomics and Fluxomics, Toulouse, France*
[3]*LabHPEC, Université de Toulouse, ENVT, Toulouse, France*
[4]*Laboratoire Hétérochimie Fondamentale et Appliquée (LHFA -UMR5069), CNRS – Université de Toulouse, Toulouse, France*
[5]*Inserm U1313, Université de Poitiers et CHU de Poitiers, Poitiers, France*
[6]*Centre de Résonance Magnétique des Systèmes Biologiques (CRMSB), Univ. Bordeaux, CNRS, CRMSB, UMR 5536, Bordeaux, France*
[7]*Institute of Metabolic and Cardiovascular Diseases, INSERM, University of Toulouse, UMR 1297, Toulouse, France*
[8]*Institut Hospitalo-Universitaire HealthAge, IHU HealthAge, Toulouse, France*
[9]*Toulouse Biotechnology Institute – INSA de Toulouse INSA/CNRS 5504 - UMR INSA/INRA 792, Toulouse, France*

Handling Editors: Paul Greenhaff & Max Petersen

The peer review history is available in the Supporting Information section of this article
(https://doi.org/10.1113/JP288871#support-information-section).

**Abstract figure legend** Using lactate tolerance tests and *in vivo* [13]C isotopic tracing experiments performed in male mice housed at different temperatures, this study demonstrates that the state of activation of non-shivering thermogenesis in brown adipose tissue (BAT) impacts systemic lactate metabolism. Cold exposure (4°C) increased blood lactate clearance, which has been associated with a higher contribution of lactate to gluconeogenesis, compared to mice housed at standard temperature (21°C). Mice housed at thermoneutrality (30°C) (as well as mice deficient for the mitochondrial uncoupling protein-1) exhibited reduced blood lactate clearance. A decreased contribution of lactate to the tricarboxylic acid cycle (TCA) as well as a reduced pyruvate cycling process fed by lactate in BAT were observed under thermoneutral

**Rémi Montané** was awarded a ministerial doctoral research scholarship to pursue postgraduate training at Toulouse University, France. The studies described in this article were conducted during his doctoral training, during which he broadly investigated the role of brown and beige adipose tissues in systemic energy metabolism. The long-term goal of Rémi's work is to contribute to a deeper understanding of metabolic maladaptation in various pathophysiological conditions.

J.C. Portais, I. Ader and A. Carriere jointly supervised this work.

conditions. This study highlights the impact of non-shivering thermogenesis on lactate clearance and metabolic fate and demonstrates the importance of considering housing temperature conditions when studying systemic lactate metabolism and inter-organ communication. Figure created using Biorender.com.

**Abstract** Non-shivering thermogenesis in brown adipose tissue (BAT) is linked to metabolic health. Yet, how its activity states impact on systemic metabolism and in particular on lactate, a highly abundant metabolite increasingly recognized as a critical player in energy metabolism, remains unresolved. The goal of this study was to investigate the impact of BAT activity on lactate metabolism at the whole organism level. To activate or inactivate non-shivering thermogenesis in BAT, we housed C57Bl6/J male mice at 4, 21 and 30°C and then conducted lactate tolerance tests. In mice exposed to cold exposure (4°C), systemic lactate clearance was elevated. In contrast, clearance of systemic lactate was poor in mice housed under thermoneutral conditions (30°C) that inactivate BAT thermogenesis, as well as in mice deficient for the mitochondrial uncoupling protein-1. To better understand lactate metabolic fate during the clearance phase, *in vivo* stable isotope tracing experiments with labelled $^{13}$C-lactate and analyses by mass spectrometry were performed. These experiments revealed that lactate contribution to gluconeogenesis was increased under cold exposure while its contribution to the tricarboxylic acid cycle was reduced in BAT under thermoneutrality. Remarkably, we also identified that lactate entered a pyruvate cycling process that was highly active in BAT, and repressed at thermoneutrality. Our study shows that inactivation of non-shivering thermogenesis decreased systemic lactate clearance, concomitantly with changes in metabolic fate of lactate in BAT and in gluconeogenic organs, in male mice.

(Received 14 March 2025; accepted after revision 2 September 2025; first published online 23 September 2025)
**Corresponding author** A. Carriere: RESTORE Research Centre, Université de Toulouse, INSERM 1301, CNRS 5070, EFS, ENVT, 4 bis avenue Hubert Curien, 31100 Toulouse, France. Email: audrey.carriere-pazat@inserm.fr

## Key points

- Lactate clearance is enhanced upon cold exposure and reduced at thermoneutrality.
- UCP1-deficient mice exhibit impaired lactate clearance.
- Oxidative utilization of lactate in brown fat is decreased at thermoneutrality.
- Prolonged cold exposure increases lactate contribution to gluconeogenesis.
- Lactate enters a highly active pyruvate cycling process in brown adipose tissue.

## Introduction

In cold environments, mammals maintain constant body temperature through thermogenesis, a metabolic process that produces heat (Lowell & Spiegelman, 2000; Ricquier, 2006). Upon acute cold exposure, muscle-related shivering thermogenesis is the main contributor for maintaining body temperature control, but with prolonged cold exposure, non-shivering thermogenesis becomes the predominant mechanism (Betz & Enerback, 2018; Foster & Frydman, 1979; Haman & Blondin, 2017). In rodents, non-shivering thermogenesis mainly occurs in brown adipose tissue (BAT), which dissipates energy as heat, and becomes inactivated when the sympathetic stimulation is reduced, that is, when stimulus (heat loss) is removed, in a state referred to as thermoneutrality (Cannon & Nedergaard, 2004; Cui

et al., 2016; Foster & Frydman, 1978; Nicholls & Locke, 1984).

BAT thermogenesis is driven by the high mitochondrial content of brown adipocytes and by the expression of the mitochondrial uncoupling protein-1 (UCP1) (Nicholls, 1976; Ricquier & Kader, 1976), which uncouples cellular respiration from ATP synthesis once activated, notably by long-chain fatty acids (Nicholls & Locke, 1984). This event strongly stimulates respiration and substrate oxidation, leading to dissipation of energy as heat at the expense of ATP synthesis. So-called beige adipocytes, which can emerge in white adipose fat pads upon stress, including cold exposure (Cousin et al., 1992; Loncar, 1991; Young et al., 1984), exhibit similar properties to brown adipocytes (Ikeda et al., 2018). While UCP1 expressed by beige adipocytes is thermogenically competent, the thermogenic capacity of beige depots

remains lower than those of the brown depot at the whole systemic level (Shabalina et al., 2013). Recently, several UCP1-independent mechanisms, including cycling of creatine, lipid and calcium, were reported to support the energy-dissipating properties of brown and beige adipocytes (Bunk et al., 2025; Ikeda et al., 2017; Kazak et al., 2015; Mottillo et al., 2014; Oeckl et al., 2022; Roesler & Kazak, 2020; Sharma et al., 2024), highlighting the complexity and versatility of metabolic pathways involved in the function of thermogenic adipose tissues.

Recently, BAT was reported to be associated with cardio-metabolic health in humans (Becher et al., 2021). Although its metabolic activity declines with age and body mass index (Cypess et al., 2009; van Marken Lichtenbelt et al., 2009; Virtanen et al., 2009), the consequences of its inactivation on whole-body metabolism have yet to be fully elucidated. In addition to high endocrine activity (Villarroya et al., 2019), BAT can regulate circulating levels of several metabolites, acting as a 'metabolic sink' notably for glucose and lipids (Fernandez-Verdejo et al., 2019; Klepac et al., 2019). While reported to regulate blood levels of succinate (Mills et al., 2018) and branched-chain amino acids (Yoneshiro et al., 2019), the impact of BAT on lactate – a highly abundant circulating metabolite recently emphasized as critical for maintaining energy homeostasis (Brooks, 2018; Rabinowitz & Enerback, 2020) – remains less well understood. While long believed to be metabolic waste, more recent studies have shown that lactate, produced when the glycolytic flux overwhelms oxidative capacities, takes on pleiotropic roles. These include regulation of redox homeostasis (Rabinowitz & Enerback, 2020), feeding biosynthesis such as gluconeogenesis (Cori & Cori, 1929) and lipogenesis (Chen et al., 2016; Katz & Wals, 1974), supporting oxidative metabolism in many organs (Hui et al., 2017), and acting as a signalling molecule, including through epigenetic modifications (Liu et al., 2023; Zhang et al., 2019). Blood lactate concentration – determined by its production and clearance – is tightly regulated (Rabinowitz & Enerback, 2020), and increased lactataemia predicts mortality in critically ill patients (Shapiro et al., 2005; Zhang & Xu, 2014). Similarly, fasting plasma lactate concentration is elevated in patients with obesity, type 2 diabetes and metabolic syndrome (Broskey et al., 2024; Lovejoy et al., 1992; Reaven et al., 1988). Although it has been recently demonstrated that lactate homeostasis is maintained through regulation of glycolysis and lipolysis (Lee et al., 2025), the mechanisms underlying altered systemic lactate metabolism in these conditions remain incompletely understood.

Recently, the relationship between lactate metabolism and brown or beige adipose tissues was investigated. Lactate has been shown to promote UCP1 expression (Barayan et al., 2023; Carriere et al., 2014; Kim et al., 2024; Liu et al., 2024; Park et al., 2019; Wang, Lee et al., 2020) and to contribute to the beiging of white adipose tissue in pathophysiological contexts such as burn (Barayan et al., 2023), cachexia (Liu et al., 2024; Sanford & Goncalves, 2024) or muscular genetic diseases (Wang, Lee et al., 2020). Both brown and beige adipocytes express high levels of the monocarboxylate transporter-1 (MCT1) on their plasma membrane (Iwanaga et al., 2009; Lagarde et al., 2021; Okamatsu-Ogura et al., 2018), mediating bidirectional lactate export and import fluxes that support glycolysis and oxidative metabolism in beige adipocytes (Lagarde et al., 2021; Petersen et al., 2017). Although glucose feeds oxidative metabolism of BAT (Jung et al., 2021; Wang, Ning et al., 2020), in rats, a large fraction of glucose consumed by brown fat was reported to be converted into lactate, particularly during severe cold exposure (Ma & Foster, 1986). However, in recent studies in mice, lactate was found to feed the tricarboxylic acid (TCA) cycle of BAT (Bornstein et al., 2023; Hui et al., 2017, 2020; Lemmer & Bartelt, 2023; Park et al., 2023) and serve as a major fuel source for thermogenesis (Park et al., 2023). These conflicting findings emphasize the need to better understand the metabolic relationships between lactate and the activity of thermogenic adipose tissues. In particular, the impact of BAT inactivation on systemic lactate clearance has not yet been addressed.

To address this knowledge gap, we performed lactate tolerance tests in male mice exhibiting inactivated or activated brown/beige adipose tissues. We found that systemic lactate clearance was impaired in mice housed at thermoneutrality. In contrast, we detected increased systemic lactate clearance in mice exposed to low temperatures. Although MCT1 invalidation in adipose tissues did not have any impact, systemic lactate clearance was impaired in mice with congenital invalidation of UCP1, further highlighting the impact of BAT on systemic lactate clearance. To better understand lactate metabolic fate during the clearance phase and according to their activation state, we performed *in vivo* isotope-tracing experiments of uniformly labelled [13]C-lactate in the same experimental conditions as for the lactate tolerance tests and analysed the [13]C-labelling pattern by mass spectrometry in brown and beige adipose tissues. We found that lactate feeds the oxidative metabolism and a highly active pyruvate cycling process particularly in BAT, which was severely repressed at thermoneutrality. We also found a greater contribution of lactate to gluconeogenesis in mice that had adapted to prolonged cold exposure. Taken together, our findings demonstrate that BAT inactivation impairs systemic lactate clearance, this being associated with modification of lactate metabolic fate in this tissue as well as in gluconeogenic organs.

# Material and methods

## Ethic approval

All animal experimentation procedures were performed in agreement with the ethical committee and registered with the French Ministry of Research (2023110712191174/2018040511443066), with efforts made to minimize animal stress and suffering. The investigators understand the ethical principles under which the journal operates and their work complies with the animal ethics checklist and with *The Journal of Physiology* guidelines. For all experiments except $^{13}$C labelling tracing experiments, mice were killed by cervical dislocation without anaesthesia. For $^{13}$C labelling experiments, decapitation was performed without anaesthesia in accordance with ethical recommendations and after approval by the ethic committee. This method was the most appropriate because it enabled us to quickly harvest a large quantity of blood without anaesthesia, avoiding the probable interference of anaesthetic drugs on tissue and blood metabolic profiles (Pierozan et al., 2017). Intraperitoneal injections were performed without anaesthesia.

## Animals

For temperature housing experiments, animal studies were carried out with C57Bl6/J male mice obtained from the Envigo laboratory (Gannat, France). Two to five mice per cage were housed in a 12 h light/dark controlled environment with unlimited access to water and standard chow diet food in a pathogen-free animal facility, with nesting material (paper towel, sizzle nest) present in all experimental conditions. Mice were housed at 21°C (regular ambient temperature), exposed at 4°C during 7 days (from 15 to 16 weeks of age) or at 30°C during 2 months (from 8 to 16 weeks of age), before being subjected to lactate intraperitoneal injection or being killed, 6 h after fasting (from 08.00 to 14.00 h during the light cycle, a moderate fasting condition used to mitigate inter-individual metabolic variability) or on fed mice, as stated in the Figure legends. Tissues were dissected and processed for histology or immediately snap-frozen in liquid nitrogen and stored at −80°C.

To obtain adipocyte-specific deletion of MCT1, mice carrying floxed alleles for Mct1 [exon 5-mct1 flox/flox, generated by Cyagen, Santa Clara, CA, USA, in collaboration with Luc Pellerin (Martini et al., 2021)] were inter-crossed with adiponectin-Cre$^{ERT2}$ mice [kindly provided by Stephan Offersman (Sassmann et al., 2010)], an adipocyte selective inducible Cre-driver line. Recombination was induced in male mice aged 16 weeks old by daily intraperitoneal (75 µl) administration of 1.5 mg tamoxifen (T5648, Sigma-Aldrich, St Louis, MO, USA, dissolved in corn oil solution) for four consecutive days. A wash-out period of 10 days was allowed before starting the experiments. Tamoxifen-treated Cre-negative littermates were used as control for experiments. The sequences of primers used for genotyping are given in Table 1.

C57BL/6J UCP1-deficient mice were kindly provided by Professor Kozak and described in Enerback et al. (1997). Male wild-type and Ucp1 knockout (*Ucp1*$^{KO}$) littermates, aged 2–5 months, housed at 21 or at 30°C for 2 months as stated in the Figure legends, were used. Primers used for genotyping are detailed in Table 1.

## Lactate tolerance test

Sixteen-week-old male mice were injected intra-peritoneally with a solution of sodium L-lactate (Sigma-Aldrich, L7022) at a dose of 6 µmol/g of body weight after a 6 h of fasting period (from 08.00 to 14.00 h during the light cycle) or on fed mice as stated in the Figure legends. Lactate was dissolved in PBS at a concentration of 1 M (osmolarity: ∼2215 mOsm, measured with a ROEBLING 13/13DR-autocal osmometer), filtered under a sterile hood and stored at −20°C until use. Control mice were injected with an iso-osmolar NaCl solution (Sigma-Aldrich, S7653) at a dose of 6 µmol/g of body weight. NaCl was dissolved in PBS at a concentration of 1 M (osmolarity: ∼2127 mOsm, measured with a ROEBLING 13/13DR-autocal osmometer). PBS-injected mice were used as a control. Blood lactate concentration was measured using the Lactate Pro 2 reader (Arkray, Minneapolis, MN, USA) from a drop of blood collected from the tail. Measurements were made before injection ($t = 0$) and every 5 min for 30 min after injection. Lactate tolerance tests were done without anaesthesia and at 21°C. We chose this experimental condition to avoid the stress that could have been generated by moving cages every 5 min from their housing places to the experimental zone. For the mode of injection, we chose intraperitoneal injection which can be quickly completed, limiting the stress for the animals, and which can be performed in alert animals, therefore avoiding anaesthesia that can hamper both lactate metabolism (Horn & Klein, 2010) and BAT activity (Ohlson et al., 1994, 2003). We acknowledge that the mode of injection can have consequences on lactate appearance (Haugen et al., 2020) and that interpretation of the results must be done in the context of these intraperitoneal injections.

## $^{13}$C-lactate *in vivo* isotope tracing

The $^{13}$C-lactate labelling experiments were performed in conditions similar to the lactate tolerance tests, with the unique exception that unlabelled sodium lactate was

**Table 1. Primer sequences**

| Gene | Forward | Reverse |
|---|---|---|
| **RTqPCR** | | |
| *18s* | AGTCCCTGCCCTTTGTACACA | CGATCCGAGGGCCTCACTA |
| *36b4* | AGTCGGAGGAATCAGATGAGGAT | GGCTGACTTGGTTGCTTTGG |
| *Ldha* | ATGCACCCGCCTAAGGTTCTT | TGCCTACGAGGTGATCAAGCT |
| *Ldhb* | CTGACCTCATCGAGTCCATG | GAAGACTTCATTCTCAATGCC |
| *Mct1* | TGTGGGCTTGGTGACCAT | AAGAGATAGATACCCGCGATGATG |
| *Mct2* | CACCACCTCCAGTCAGATCG | CTCCCACTATCACCACAGGC |
| *Mct4* | AGTGCCATTGGTCTCGTG | CATACTTGTTAAACTTTGGTTGCATC |
| *Me1* | GTCGTGCATCTCTCACAGAAG | TGAGGGCAGTTGGTTTTATCTTT |
| *Pkm1* | GCTGTTTGAAGAGCTTGTGC | TTATAAGAGGCCTCCACGCT |
| *Pepck1* | ATGTTCGGGCGGATTGAAG | TCAGGTTCAAGGCGTTTTCC |
| *Mdh1* | AGGAAGGACCTACTGAAAGC | GAGTAGAGCAGGTCATCAGG |
| *Mdh2* | GCAAGATCACTCCTTTTGAG | AAGAGATGCTGATGCTGACT |
| *Pc* | GCCCAGAAGTTGCTACATTACCT | CTCACATTGACAGGGATTGGA |
| *Ucp1* | GACCGACGGCCTTTTTCAA | AAAGCACACAAACATGATGACGTT |
| *Cidea* | GCCGTGTTAAGGAATCTGCTG | TGCTCTTCTGTATCGCCCAGT |
| *Pgc1a* | AAAGGATGCGCTCTCGTTCA | GGAATATGGTGATCGGGAACA |
| **Genotyping** | | |
| *CRE[ERT2]* | TGGTGCATCTGAAGACACTACA | TGCTGTTGGATGGTCTTCACAG |
| *MCT1LoxP* | AGACTTGGGTAACTGAATGATGCTGACT | TCCAAGGACAGCCAAGCTACATAGAG |
| *Ucp1-Neo* | GGTAGTATGCAAGAGAGGTGT | CCTACCCGCTTGCATTGCTCA |
| *Ucp1-E2* | GGTAGTATGCAAGAGAGGTGT | CCTAATGGTACTGGAAGCCTG |

replaced by sodium L-[U-$^{13}$C]-lactate (Sigma-Aldrich, 485 926). A total of five mice per group were used for $^{13}$C-lactate injection and two control mice injected with unlabelled sodium L-lactate were used to ensure proper identification of labelled signals in the mass spectrometry spectra. Fifteen minutes after injection, organs were snap-frozen in liquid nitrogen and stored at −80°C until metabolite extraction. Plasma and tissue samples were collected and snap-frozen within 5 min to prevent non-physiological changes in metabolite content.

### Locomotion test

Mice locomotion was evaluated by placing individual 16-week-old male mice in Digital Ventilated Cages (DVC, Tecniplast) with unlimited access to water, standard chow diet food and nesting material. Mice were acclimated 12 h before the test, after which average speed and distance were measured during 30 min after intraperitoneal injection of sodium L-lactate and iso-osmolar NaCl solution (6 µmol/g of body weight, with 1 M solutions as for the intraperitoneal lactate tolerance test) in mice fasted for 6 h. PBS-injected mice were used as a control.

### Extraction of polar metabolites from plasma samples

Polar metabolites were extracted from plasma samples by adding 1 mL of cold MeOH/H$_2$O (4:1) to 20 µL of plasma. After thorough mixing and incubation at

−20°C for 15 min, samples were centrifuged (16,000 *g*, 15 min, 4°C). The supernatant was collected, dried using a miVac Concentrator (Genevac Inc, New York, NY, USA) and resuspended into 150 µL acetonitrile/water (3:1) supplemented with 15 mM ammonium acetate. The extract was transferred to HPLC vials for analysis.

### Extraction of polar metabolites from organs and tissues

Frozen tissues were first freeze-dried and ground to powder using a Mixer Mill MM 400 (Retsch GmBH, Haan, Germany) operated with dry ice. Then, 5 mg of the homogenized powder was mixed with 2 mL of cold methanol/acetonitrile/water (2:2:1) solution containing 0.1% formic acid, followed by vortexing for 2 min, incubation at −20°C for 25 min, and centrifugation (16,000 *g*, 15 min, 4°C). The supernatant was collected, dried and resuspended into 125 µL acetonitrile/water (3:1) supplemented with 15 mM ammonium acetate. The extract was transferred to HPLC vials for analysis.

### Analysis of metabolite labelling by HPLC/HRMS

The HPLC/high-resolution MS (HRMS) analyses were performed using an UHPLC Vanquish FLEX chromatographic system coupled to an Orbitrap Q Exactive+ mass spectrometer (Thermo Fisher Scientific,

Waltham, MA, USA) operated in negative ion mode (ESI−). Metabolites were separated on a P120 HILIC-Z (2.1 × 150 mm i.d., 2.7 μm) column (Agilent, Santa Clara, CA, USA) at 30°C. The mobile phases of the HILIC method consisted of A, acetonitrile/$H_2O$ (9:1) and B, acetonitrile/$H_2O$ (1:9). Both A and B phases were supplemented with 15 mM ammonium acetate. HILIC separation was performed at 250 μL/min with the following gradient (min, %B): 0, 15%; 4, 25%; 5.5, 30%; 13.5, 35%; 15.9, 50%. The column was then equilibrated for 10 min at the initial conditions before the next sample was analysed. The injection volume was 2–5 μL.

The HRMS analyses were performed in full-scan mode with a resolution of 70,000 (at 400 *m/z*) over the *m/z* range 60–1000. The automatic gain control (AGC) target was set to $1^{e6}$ with a maximum injection time of 100 ms. Data were acquired with the following source parameters: the capillary temperature was 250°C, the source heater temperature, 300°C, the sheath gas flow rate, 45 a.u. (arbitrary units), the auxiliary gas flow rate, 10 a.u., the sweep gas flow rate, 1.0 a.u., the S-Lens RF level, 55 %, and the source voltage, 2.70 kV (ESI− mode). Data were acquired in a single analytical batch. Biological samples were randomized in the analytical run and control quality (QC) samples, consisting of a mixture of all biological samples of the same type, that is, plasma or tissue samples, were injected at regular intervals throughout the experiment. A mixture of metabolite standards was analysed to determine the retention time corresponding to each compound in biological samples. Metabolites were then identified by extracting the accurate mass with a tolerance of 5 ppm. The raw MS isotopic profiles of metabolites were then processed using TraceFinder (Thermo Fisher Scientific). The isotopologue fractions were obtained after correcting for natural isotopic abundances using IsoCor (https://github.com/MetaSys-LISBP/IsoCor). The molecular $^{13}C$ enrichments (Table S1) were calculated from the sum of the relative abundances of all isotopologues of a metabolite weighted by the number of $^{13}C$ atoms in each isotopologue. Normalized molecular $^{13}C$ enrichments in metabolites within the different tissues or within the plasma were calculated by normalizing to the molecular $^{13}C$ enrichment of lactate within the different tissues or within the plasma, respectively, for the same animal.

## Histological studies

For light microscopy, tissues were fixed for 24 h in 10% buffered formalin before stored in PBS at 4°C. After paraffin embedding, 3 μm thick paraffin sections were dewaxed (successive toluene and descending alcohol baths). Haematoxylin and Eosin (HE) stain was used. The stained sections were observed using a Nikon Eclipse Ci microscope and then scanned (Panoramic Desk, 3D Histec). Microscopic evaluation and photomicrograph captures were carried out using CaseViewer 2.4.0 software.

## Immunofluorescence microscopy

Interscapular BAT (iBAT) was fixed in 10% buffered formalin overnight before being embedded in 2% agarose and cut into 300 μm sections using a vibratome (Campden). Sections were incubated in blocking solution (2% normal horse serum and 0.2% triton X-100 in PBS) at room temperature for 6 h before being incubated for 24 h at room temperature with primary antibodies (chicken anti-MCT1 1:100 EMD Millipore AB1286I, Billerica, MA, USA). After overnight incubation at 4°C with Alexa557-conjugated secondary antibody (anti-chicken Life Technology, NL016, 1:200, Carlsbad, CA, USA) and a 30 min incubation with DAPI (1:10,000), imaging was performed using a confocal laser scanning microscope [LSM880 (Zen Blue v2.3-2), Carl Zeiss, Oberkochen, Germany], and image analysis was conducted using Fiji software (v2.1.0-1, National Institutes of Health, Bethesda, MD, USA).

## RNA extraction and real-time PCR

Total RNA from tissues were extracted with Qiazol and isolated using RNeasy minicolumns (Qiagen, Valencia, CA, USA). For relative real-time PCR analysis, 1000 ng of total RNA was reverse transcribed using the High Capacity cDNA Reverse Transcription kit (Life Technologies/Applied Biosystems; 4 368 814). Gene expression was determined with cDNA reversed transcripts, SYBR Green PCR Master Mix (Life Technologies/Applied Biosystem; 4 309 155) and 300 nmol/L primers (Eurogentec, Seraing, Belgium) on an Applied Biosystem QS5 instrument. Primers are listed in Table 1. Relative gene expression was determined using the $2^{-\Delta\Delta CT}$ method as indicated and normalized to the geometrical mean of the *36b4* and *18s* housekeeping genes.

## Statistical analysis

Data are expressed as mean ± SD after outlier exclusion with 1% Rout correction. Graphic representation of the data as well as all statistical tests were performed with GraphPad Prism. Data were tested for normal distribution using the Shapiro–Wilk test. When the distribution was normal, a parametric test was used. When data did not show a normal distribution, data were log-transformed and tested for normality. If the transformed data showed a normal distribution, we performed parametric tests on the transformed data. If

transformed data did not show a normal distribution, we performed non-parametric tests on the non-transformed data. Comparison between three groups was done using one-way ANOVA with Welch's correction if variances were not equal between groups, and with Tukey's or Dunnett's T3 multiple comparisons tests when variances were equal or not equal, respectively. When data did not show a normal distribution, a non-parametric Kruskal–Wallis test with Dunn's multiple comparison test was performed for comparison between three groups. For lactate tolerance tests, a two-way ANOVA with Tukey's multiple comparisons test was performed. Comparison between two groups was performed with a Student's *t* test with Welch's correction if variances were not equal between groups or with a Mann–Whitney test. *N* values are specified in the Figure legends. Exact *P* values are reported on the Figure legends. Differences were considered statistically significant at $P \leq 0.05$.

## Results

### Systemic clearance of lactate changes drastically with housing temperature of mice

We first assessed the impact of housing temperature on systemic lactate clearance. To this end, we acclimatized 4-month-old male mice to prolonged cold (exposure to 4°C over 1 week; '4°C mice'), a condition known to strongly activate BAT thermogenesis. We also housed groups of mice at two different temperatures: standard animal facility temperature, known to mildly activate BAT thermogenesis (21°C, referred to as '21°C mice'), and thermoneutrality, known to inactivate BAT thermogenesis (30°C for 8 weeks, referred to as '30°C mice'). We then conducted lactate tolerance tests, in fed animals. After sodium L-lactate injection, the blood lactate concentration increased in 21°C mice, reaching 10 mM as soon as 5 min after the injection (mimicking the physiological response to physical exercise; Goodwin et al., 2007), followed by a gradual return to basal levels after 30 min (Fig. 1*A*), a profile similar to one recently published (Moberg et al., 2024). Relative to 21°C mice, lactate clearance was elevated in 4°C mice and lowered in 30°C mice such that blood lactate levels did not return to basal levels 30 min after the injection (Fig. 1*A*). These different responses reached statistical significance in comparative analysis of the areas under the curve between 4, 21 and 30°C mice (Fig. 1*B*). In 6 h fasted conditions (during the light cycle – from 08.00 to 14.00 h – a moderate fasting condition used to mitigate inter-individual metabolic variability), we noted a similar profile (Fig. 1*C*, *D*). Although a state of malaise in mice injected with sodium L-lactate has been reported with hypertonic solutions (Lund et al., 2023), we did not detect any differences in locomotion activity between control or experimental groups in our conditions of injection

(Fig. 1*E*, *F*). In addition, neither PBS nor NaCl injection had significant impact on blood lactataemia, excluding the possibility that rising blood lactate levels following sodium L-lactate injection are caused by changes in osmolarity (Fig. 1*G*).

While we did not detect any statistically significant differences in body weight across experimental groups (Fig. 1*H*), iBAT weight was significantly higher in 30°C mice (Fig. 1*I*, *J*) while no change was observed for the subcutaneous adipose tissue (SCAT) (Fig. 1*K*, *L*). As expected, lipid droplet size was dramatically reduced in iBAT and SCAT from 4°C mice compared to 21°C mice, while lipid droplets were enlarged in 30°C mice, especially in iBAT (Fig. 1*M*). mRNA levels of Ucp1 in iBAT were significantly elevated in 4°C mice compared to 21°C mice and decreased in 30°C mice which also exhibited reduced expression of *Pgc1α* and *Cidea* (Fig. 1*N–P*). A similar gene expression profile was observed in SCAT (Fig. 1*Q–S*). These findings validate expected changes in iBAT and SCAT according to mouse housing temperatures. Taken together, these results demonstrate that clearance of systemic lactate is increased in mice acclimatized to prolonged cold and decreased under thermoneutrality.

### UCP1 knockout mice exhibit reduced lactate clearance

We next assessed the consequence of *Ucp1* genetic deficiency on systemic lactate clearance. Here, we challenged *Ucp1*-deficient (*Ucp1*[KO]) mice and their wild-type littermates – housed at 21°C – with a lactate tolerance test. We found that *Ucp1*[KO] mice exhibited reduced lactate clearance with the presence of a plateau at the later time points of the kinetic experiment (Fig. 2*A*), which translated into a statistically significant increased area under the curve compared to littermate controls (Fig. 2*B*). Housing *Ucp1*[KO] mice at 30°C did not further increase lactate intolerance (Fig. 2*C*, *D*), suggesting that both thermoneutral housing and *Ucp1*[KO] mice models share common mechanism(s) that contribute to the regulation of systemic lactate clearance, one of which may be linked to *Ucp1* downregulation. These data reveal that UCP1 impacts systemic lactate metabolism and further highlight, using a genetic model, the impact of thermogenic adipose tissues on systemic lactate clearance.

### Remodelling of lactate-related gene expression in brown and beige adipose tissues in different thermogenic states

We then investigated whether the different thermogenic states were associated with remodelling of lactate-related gene expression in thermogenic adipose tissues. We found that mRNA levels of *Ldha* and *Ldhb* – which encode enzymes involved in the reversible conversion of pyruvate

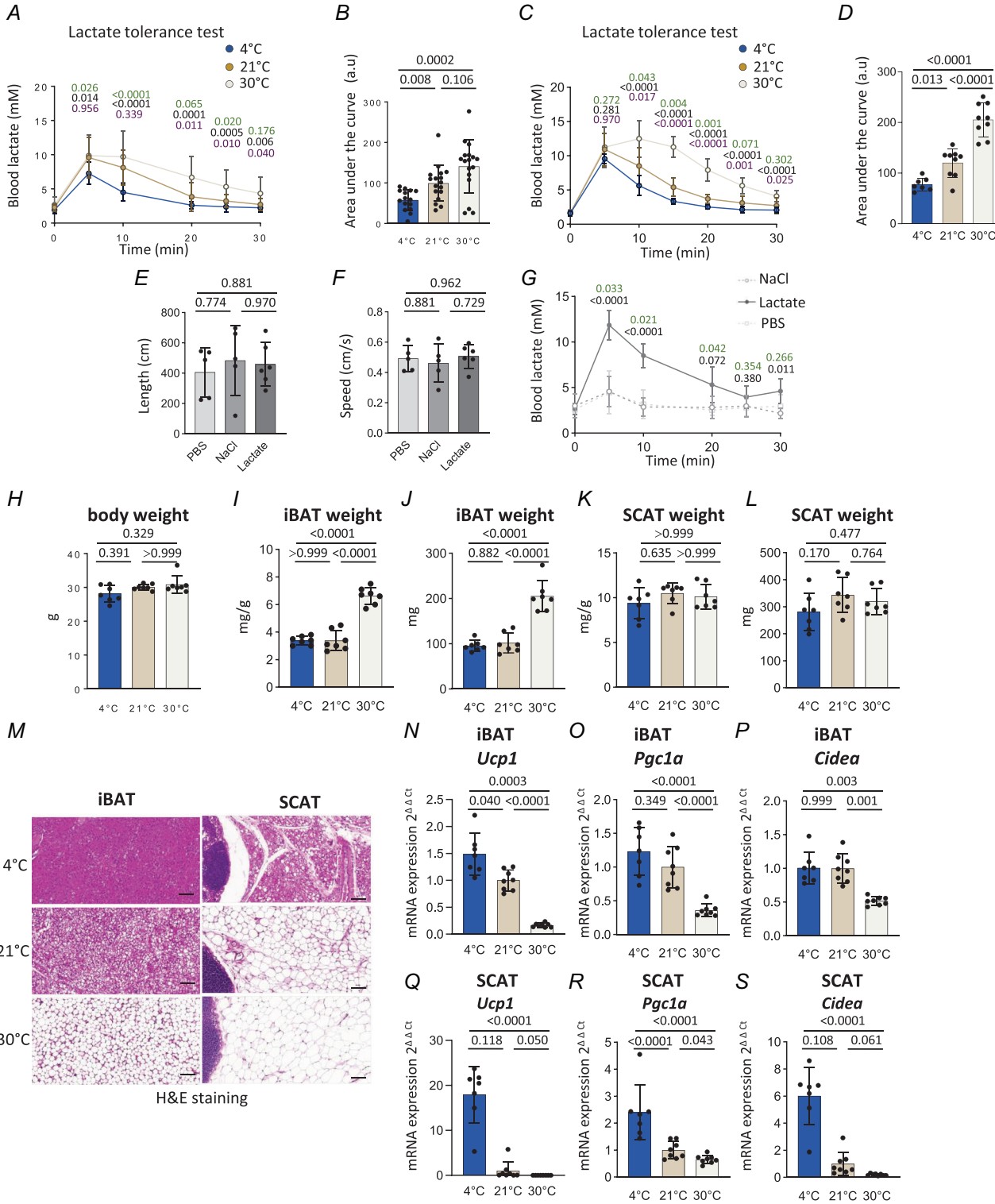

**Figure 1. Systemic lactate clearance is increased in mice acclimatized to prolonged cold and decreased at thermoneutrality**

*A*, intraperitoneal lactate tolerance tests on fed mice previously exposed at 4°C or housed at 21 or 30°C (*n* = 15–17 per group; 4°C *vs.* 21°C in green, 4°C *vs.* 30°C in black, 21°C *vs.* 30°C in violet, two-way ANOVA). *B*, area under the curve of intraperitoneal lactate tolerance tests shown in *A* (*n* = 15–17 per group, one-way ANOVA). *C*, intraperitoneal lactate tolerance tests on 6 h fasted mice previously exposed at 4°C or housed at 21 or 30°C (*n* = 7–9 per group; 4°C *vs.* 21°C in green, 4°C *vs.* 30°C in black, 21°C *vs.* 30°C in violet, two-way ANOVA). *D*,

area under the curve of intraperitoneal lactate tolerance tests shown in C ($n = 7$–9 per group, one-way ANOVA). E, distance moved in response to injections (PBS, NaCl or sodium L-lactate) of 6 h fasted mice housed at 21°C ($n = 5$–6 per group, one-way ANOVA). F, movement speed in response to injections (PBS, NaCl or sodium L-lactate) of 6 h fasted mice housed at 21°C ($n = 5$–6 per group, one-way ANOVA). G, blood lactate levels in response to injections (PBS, NaCl or sodium L-lactate) of fed mice housed at 21°C ($n = 3$–6 per group, lactate *vs*. PBS in green, lactate *vs*. NaCl in black, two-way ANOVA). H, body weight (g) of 6 h fasted mice previously exposed at 4°C or housed at 21 or 30°C ($n = 7$ per group, Kruskal-Wallis). I–L, weight of iBAT [I (mg/g body weight), J (mg)] and SCAT [K (mg/g body weight), L (mg)] of 6 h fasted mice previously exposed at 4°C or housed at 21 or 30°C [$n = 7$ per group, one-way ANOVA (I, J, L) and Kruskal–Wallis (K)]. M, representative histological images of haematoxylin and eosin stained slides of iBAT and SCAT harvested from 6 h fasted mice previously exposed at 4°C or housed at 21 or 30°C ($n = 8$ per group, scale bar 100 µm). N–P, mRNA levels of *Ucp1* (N), *Pgc1α* (O) and *Cidea* (P) in iBAT of 6 h fasted mice previously exposed at 4°C or housed at 21 or 30°C ($n = 8$ per group, one-way ANOVA). Q–S, mRNA levels of *Ucp1* (Q), *Pgc1α* (R) and *Cidea* (S) in SCAT of 6 h fasted mice previously exposed at 4°C or housed at 21 or 30°C [$n = 8$ per group, Kruskal–Wallis (Q, S) and one-way ANOVA (R)]. Mice were 4-month-old males. Data are presented as mean ± SD.

into lactate – as well as those of the lactate transporter *Mct1* – were very sensitive to thermogenic conditions. Specifically, we detected statistically significant higher gene expression of *Ldha*, *Ldhb* and *Mct1* in iBAT from 4°C mice relative to 30°C mice (Fig. 3A–C). MCT2 and MCT4, isoforms of monocarboxylate transporters, also transport monocarboxylates such as lactate. *Mct2*

expression was significantly downregulated in iBAT of 4°C mice relative to 21°C mice, but there was no difference between 21 and 30°C mice (Fig. 3D) while *Mct4* expression increased in 30°C mice although with a high inter-individual variability (Fig. 3E). In SCAT, similar trends were observed for *Ldha*, *Ldhb* and *Mct1* (Fig. 3F–J). Together, these results indicate that different thermogenic

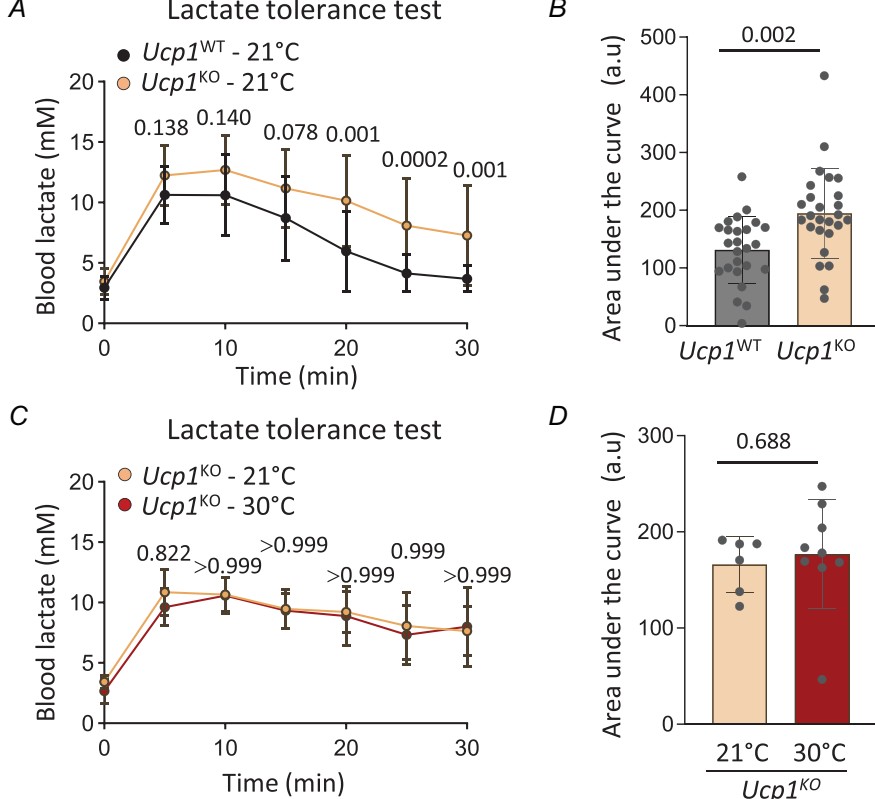

**Figure 2. UCP1$^{KO}$ mice exhibit reduced lactate clearance**
A, intraperitoneal lactate tolerance tests on *Ucp1*$^{WT}$ and *Ucp1*$^{KO}$ fed mice housed at 21°C ($n = 25$–26 per group, two-way ANOVA). B, area under the curve of intraperitoneal lactate tolerance tests shown in A ($n = 25$–26 per group, two-tailed Student's t test). C, intraperitoneal lactate tolerance tests on *Ucp1*$^{KO}$ fed mice housed at 21 or 30°C ($n = 6$–9 per group, two-way ANOVA). D, area under the curve of intraperitoneal lactate tolerance tests shown in A ($n = 6$–9 per group, two-tailed Student's t test). Mice were 2- to 5-month-old males. Data are presented as mean ± SD.

conditions induced a remodelling of lactate-related gene expression in iBAT and SCAT.

## Adipose MCT1 deficiency does not impact clearance of systemic lactate

As our results indicated reduced expression of the lactate transporter *Mct1* in iBAT and SCAT in 30°C mice (Fig. 3*C*, *H*) and as have we previously shown the role of MCT1 in lactate fluxes in thermogenic adipocytes (Lagarde et al., 2021), we next investigated whether reduction of MCT1 expression in adipose tissues would mimic effects on lactate clearance observed with thermoneutral housing. To answer this question, we intercrossed Mct1$^{flox/flox}$ mice (described in Martini et al., 2021) with adiponectin Cre$^{ERT2}$ mice (described in Sassmann et al., 2010) to generate control (*Mct1*$^{WT}$) and adipose tissue-specific *Mct1* knockout mice (*Mct1*$^{\Delta Ad}$) (Fig. 4*A*). Two weeks after the first tamoxifen injection, *Mct1* mRNA levels in iBAT and SCAT were significantly reduced in *Mct1*$^{\Delta Ad}$ mice compared to *Mct1*$^{WT}$ mice while no modification was observed for both *Mct2* and *Mct4* expression (Fig. 4*B–G*).

In addition, MCT1 protein expression in adipocytes was strongly reduced, as demonstrated by MCT1 immuno-fluorescence staining analysis (Fig. 4*H*). Despite MCT1 reduction in adipose tissues, the lactate tolerance profiles of *Mct1*$^{\Delta Ad}$ mice and *Mct1*$^{WT}$ mice were similar (Fig. 4*I*, *J*), suggesting that MCT1 expressed in iBAT and SCAT adipocytes does not contribute to systemic clearance of blood lactate.

## Cold exposure increases lactate contribution to gluconeogenesis while thermoneutral housing decreases lactate contribution to the TCA cycle in BAT

We next aimed to characterize lactate metabolic fate in brown and beige adipose tissues during the clearance phase, especially at the time when blood lactate concentration was decreasing. For this, we performed lactate tolerance tests on 4, 21 and 30°C mice in exactly the same conditions as before, except that unlabelled lactate was replaced by $^{13}$C-labelled lactate. Plasma and tissues were then collected 15 min after the injection, that is, when blood lactate level was decreasing in all

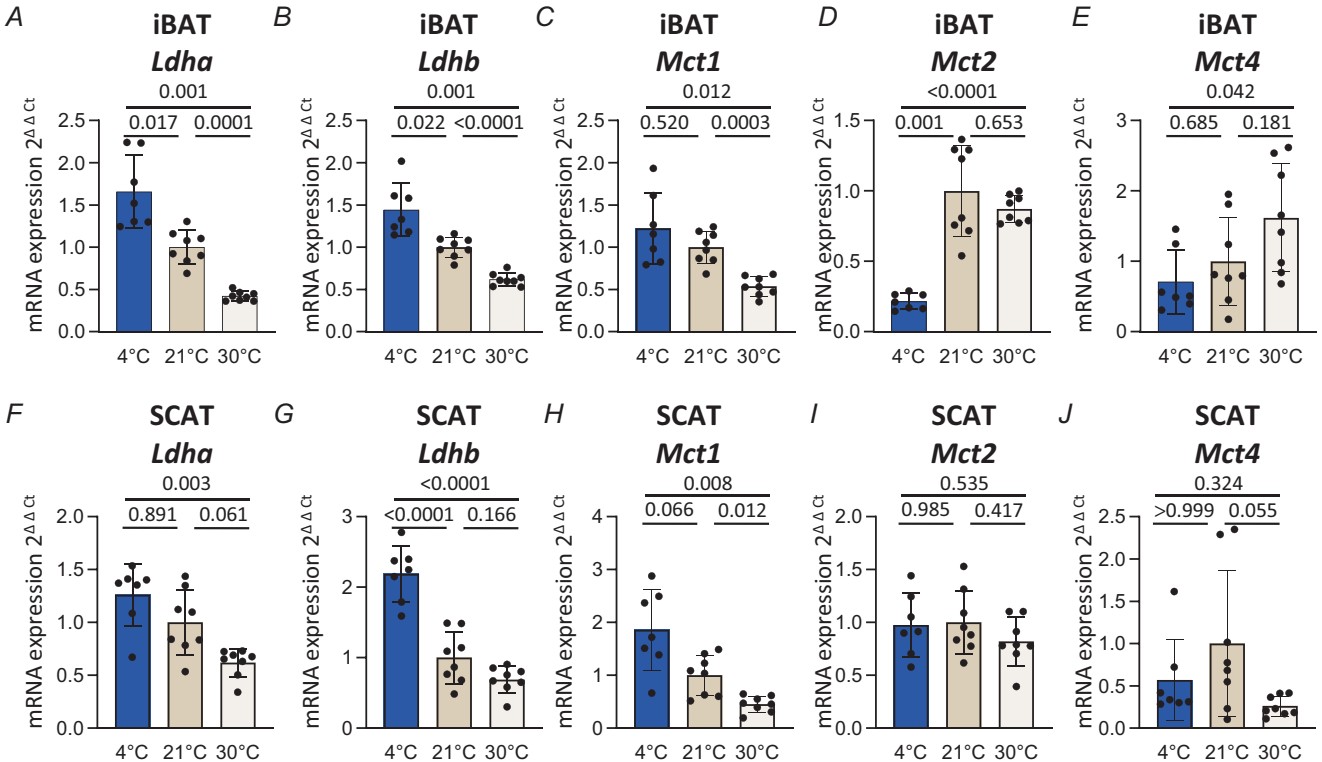

**Figure 3. Housing temperatures remodel lactate-related gene expression in thermogenic adipose tissues**
*A–E*, mRNA levels of *Ldha* (*A*), *Ldhb* (*B*), *Mct1* (*C*), *Mct2* (*D*) and *Mct4* (*E*), in iBAT of 6 h fasted mice previously exposed at 4°C or housed at 21 or 30°C (*n* = 7–8 per group, one-way ANOVA). *F–J*, mRNA levels of *Ldha* (*F*), *Ldhb* (*G*), *Mct1* (*H*), *Mct2* (*I*) and *Mct4* (*J*), in SCAT of 6 h fasted mice previously exposed at 4°C or housed at 21 or 30°C [*n* = 7–8 per group, Kruskal–Wallis (*F*, *J*) and one-way ANOVA (*G–I*)]. Mice were 4-month-old males. Data are presented as mean ± SD.

conditions but the highest differences were observed between the three groups (Fig. 1*C*). After metabolite extraction, we analysed $^{13}$C labelling in lactate and downstream metabolites by LC-HRMS. The recovery of $^{13}$C atoms in circulating lactate was around 14% on average (Fig. 5*A*), indicating that the $^{13}$C-lactate injected at $t = 0$ was significantly diluted by $^{12}$C-lactate endogenously produced by all tissues of the organism 15 min after the injection. No statistically significant differences in the mean $^{13}$C enrichment of lactate in plasma, iBAT and SCAT were observed between the different thermogenic conditions (Fig. 5*A*). We then characterized the $^{13}$C labelling in downstream metabolites. As tissues can use both labelled and unlabelled (endogenous) lactate, the contribution of the labelled lactate to the production of different metabolites requires the normalization of the $^{13}$C enrichment of the different metabolites to the $^{13}$C enrichment of the source. Because $^{13}$C-lactate was the only $^{13}$C source, the relative contribution from the tracer to downstream metabolites was calculated by dividing the enrichment of the metabolite with the enrichment of the tracer as performed in other studies (Cai et al., 2025; Jimenez-Blasco et al., 2024; Wang, Kwon et al., 2020; Wang, Ning et al., 2020). We did not observe any impact of the different thermogenic conditions on the relative contribution of lactate to pyruvate in plasma,

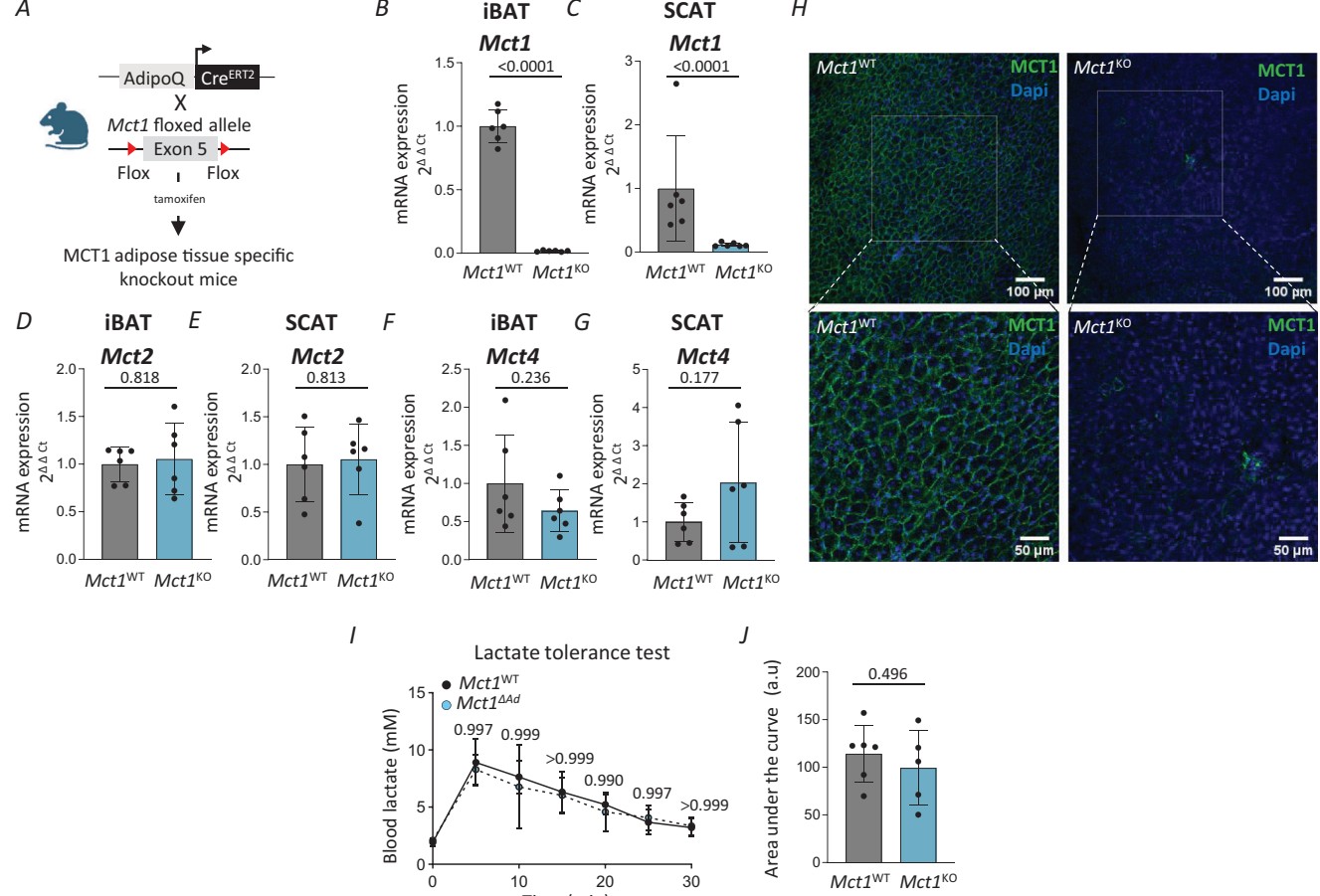

**Figure 4. Adipose MCT1 deficiency does not impact systemic lactate clearance**
*A*, schematics illustrating the adipose-specific MCT1 deficiency mice model. *B* and *C*, mRNA levels of *Mct1* on iBAT (*B*) and SCAT (*C*) of *Mct1*$^{WT}$ and *Mct1*$^{\Delta Ad}$ 6 h fasted mice housed at 21°C (*n* = 6 per group, two-tailed Student's *t* test). *D* and *E*, mRNA levels of *Mct2* on iBAT (*D*) and SCAT (*E*) of *Mct1*$^{WT}$ and *Mct1*$^{\Delta Ad}$ 6 h fasted mice housed at 21°C [*n* = 6 per group, two-tailed Mann–Whitney (*D*) and Student's *t* test (*E*)]. *F* and *G*, mRNA levels of *Mct4* on iBAT (*F*) and SCAT (*G*) of *Mct1*$^{WT}$ and *Mct1*$^{\Delta Ad}$ 6 h fasted mice housed at 21°C (*n* = 6 per group, two-tailed Student's *t* test). *H*, representative confocal images of 300 µm sections of iBAT from 6 h fasted *Mct1*$^{WT}$ and *Mct1*$^{\Delta Ad}$ mice housed at 21°C, stained with an antibody recognizing MCT1 (green). Nuclei were stained with DAPI (blue). Top images: 10× magnification; scale bar 100 µm (*n* = 3 per group). Bottom images: cropped and 2× zoomed-in sections of top images (scale bar 50 µm). *I*, intraperitoneal lactate tolerance test on 6 h fasted *Mct1*$^{WT}$ and *Mct1*$^{\Delta Ad}$ mice housed at 21°C (*n* = 5–6 per group, two-way ANOVA). *J*, area under the curve of intraperitoneal lactate tolerance tests shown in *I* (*n* = 5–6 per group, two-tailed Student's *t* test). Mice were 4-month-old males. Data are presented as mean ± SD.

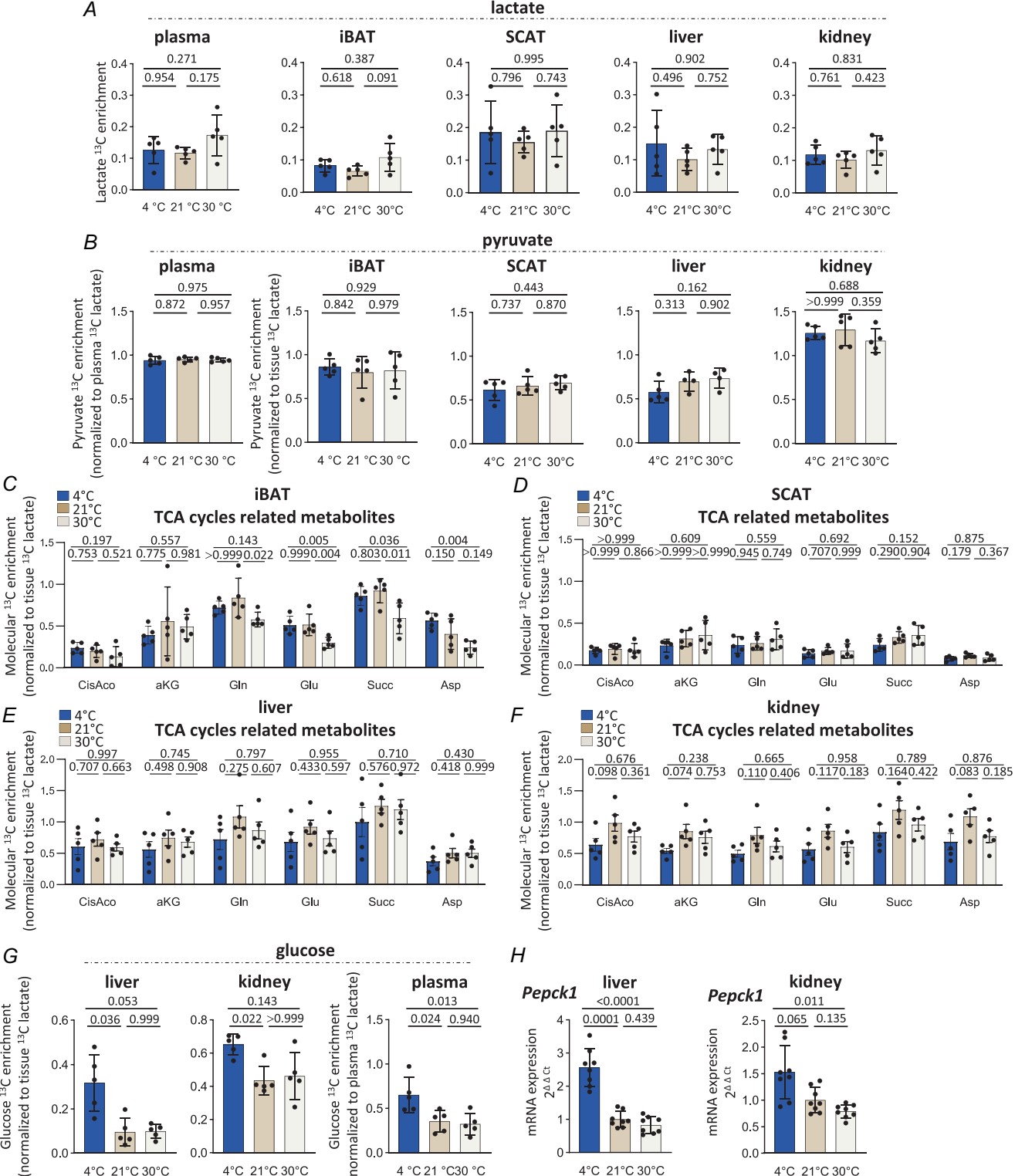

**Figure 5. Cold exposure increases lactate contribution to gluconeogenesis while thermoneutral housing decreases lactate contribution to the TCA cycle in brown adipose tissue**

*A*, molecular $^{13}C$ enrichment of lactate in plasma and tissues (iBAT, SCAT, liver and kidney) 15 min after intra-peritoneal injection of [U-$^{13}C$]-lactate to 6 h fasted mice previously exposed at 4°C or housed at 21 or 30°C [*n* = 5 per group, one-way ANOVA (plasma, iBAT, SCAT, liver and kidney)]. *B*, ratio of the plasma or tissue (iBAT, SCAT, liver and kidney) molecular $^{13}C$ enrichment of pyruvate to the plasma or tissue (iBAT, SCAT, liver and kidney) molecular $^{13}C$ enrichment of lactate respectively 15 min after intraperitoneal injection of [U-$^{13}C$]-lactate to 6 h fasted mice

previously exposed at 4°C or housed at 21 or 30°C [$n = 4–5$ per group, one-way ANOVA (plasma, iBAT, SCAT, liver) and Kruskal–Wallis (kidney)]. C–F, ratio of the tissue molecular $^{13}$C enrichment of TCA cycle-related metabolites (CisAco, cis-aconitate; aKG, $\alpha$-ketoglutarate, Gln, glutamine; Glu, glutamate, Succ, succinate; and Asp, aspartate) to the tissue molecular $^{13}$C enrichment of lactate in iBAT (C), SCAT (D), liver (E) and kidney (F) 15 min after intra-peritoneal injection of [U-$^{13}$C]-lactate to 6 h fasted mice previously exposed at 4°C or housed at 21 or 30°C ($n = 5$ per group). iBAT: one-way ANOVA (CisAco, aKG, Glu, Succ, Asp) or Kruskal–Wallis (Gln). SCAT: one-way ANOVA (Gln, Glu, Succ, Asp) or Kruskal–Wallis (CisAco, aKG). Liver: one-way ANOVA (CisAco, aKG, Gln, Glu, Succ, Asp). Kidney: one-way ANOVA (CisAco, aKG, Gln, Glu, Succ, Asp). G, ratio of the plasma or tissue (liver and kidney) molecular $^{13}$C enrichment of glucose to the plasma or tissue (liver and kidney) molecular $^{13}$C enrichment of lactate respectively after intraperitoneal injection of [U-$^{13}$C]-lactate to 6 h fasted mice previously exposed at 4°C or housed at 21 or 30°C [$n = 5$ per group, one-way ANOVA (liver, plasma) and Kruskal–Wallis (kidney)]. H, mRNA levels of Pepck1 in liver and kidney of 6 h fasted mice previously exposed at 4°C or housed at 21 or 30°C ($n = 7–8$ per group, one-way ANOVA). Mice were 4-month-old males. Data are presented as mean ± SD.

iBAT and SCAT (Fig. 5B). Next, we examined how the $^{13}$C-labelling profile of TCA cycle-related metabolites was impacted by housing temperatures. Here, although no differences were detected between 4 and 21°C mice, we found that housing mice at 30°C significantly decreased the relative contribution of lactate to glutamine (Gln), glutamate (Glu), succinate (Succ) and aspartate (Asp) labelling in iBAT, as we observed lower normalized $^{13}$C-molecular enrichments in TCA-related metabolites in this tissue (Fig. 5C), while no changes were observed in SCAT (Fig. 5D). These data demonstrated that the relative contribution of lactate to the TCA cycle decreased in iBAT, but not in SCAT, in the thermoneutral state.

We next analysed $^{13}$C labelling in two other tissues, namely liver and kidney. Although no difference was observed for the $^{13}$C labelling rate of lactate nor for the contribution of lactate to pyruvate and TCA-related metabolites (Fig. 5A, B, E), we found a higher contribution of lactate to glucose in the liver of 4°C mice compared to 21 and 30°C mice, concomitant with a significant increase in plasma glucose $^{13}$C enrichment in 4°C mice (Fig. 5G). A similar profile was observed in the kidney (Fig. 5A, B, F, G). However, relative to standard housing conditions, thermoneutral housing did not have any impact on lactate contribution to gluconeogenesis in both tissues (Fig. 5G). These findings are consistent with the higher mRNA levels of the cytosolic gluconeogenic phosphoenolpyruvate kinase enzyme (Pepck1), in both the liver and kidney of 4°C mice only (Fig. 5H). Together, these data indicate that lactate is a key contributor for fuelling gluconeogenesis in mice adapted to prolonged cold exposure, a process that may contribute to the increased systemic lactate clearance of 4°C mice. However, as gluconeogenesis fed by lactate was not decreased in 30°C mice compared to 21°C mice, a gluconeogenesis-independent mechanism may contribute to altering lactate clearance in the thermoneutral state.

## Thermoneutral housing decreases pyruvate cycling in BAT

The major labelled isotopologue to be detected in plasma lactate was the m + 3 in all three conditions, consistent with the injection of [U-$^{13}$C]-lactate (Fig. 6A). However, we also observed m + 1 (singly labelled) and m + 2 (doubly labelled) lactate species, that is, one and two $^{13}$C carbons among the three carbon atoms of lactate species – especially in iBAT. The singly and doubly labelled species were present at high levels and even at a level similar to that of the triply labelled one in iBAT of 4°C mice (Fig. 6A). Interestingly, this particular iBAT isotopologue profile was less pronounced at 30°C, m + 3 fractions tending to be higher than m + 1 and m + 2 fractions in this condition (Fig. 6A). The lactate m + 1 and m + 2 isotopologues could be also detected in liver and kidney, though at a lower level than in iBAT, but were almost undetected in SCAT (Fig. 6A). Similar isotopologue profiles were observed for pyruvate (Fig. 6B).

The occurrence of m + 1 and m + 2 species is due to pyruvate cycling, which has been described in other tissues such as the brain (Cerdan, 2017), and is a metabolic process where pyruvate is used and re-synthesized. It occurs via mechanisms (Fig. 6C) that include: (i) the carboxylation of pyruvate into oxaloacetate via pyruvate carboxylase (causing the introduction of $^{12}$C atoms into the initially triply labelled pyruvate, Fig. 6D, left panel); (ii) full oxaloacetate rotation into the TCA cycle (causing further isotope dilution and loss of label after decarboxylation and label scrambling in oxaloacetate or malate) or the reverse equilibrium of oxaloacetate with other four-carbon intermediates of the TCA cycle (causing label scrambling in oxaloacetate or malate) (Fig. 6D, left panel); and (iii) the conversion of either oxaloacetate [via phosphoenolpyruvate carboxykinase (PEPCK) + pyruvate kinase] or malate (via the malic enzyme) by decarboxylation back into pyruvate (Fig. 6C, D). As a result, the recycled pyruvate can have lost one (m + 2 species) or two (m + 1 species) $^{13}$C atoms (Fig. 6D, left panel) as previously demonstrated (Hasenour et al., 2020). The m + 1 and m + 2 isotopologues can be also obtained if pyruvate enters the TCA cycle via pyruvate dehydrogenase activity (Fig. 6D, right panel), and is further re-formed from cytosolic malate or oxaloacetate, a process that requires additional anaplerotic activity to replenish the TCA cycle, which is mostly pyruvate carboxylase in

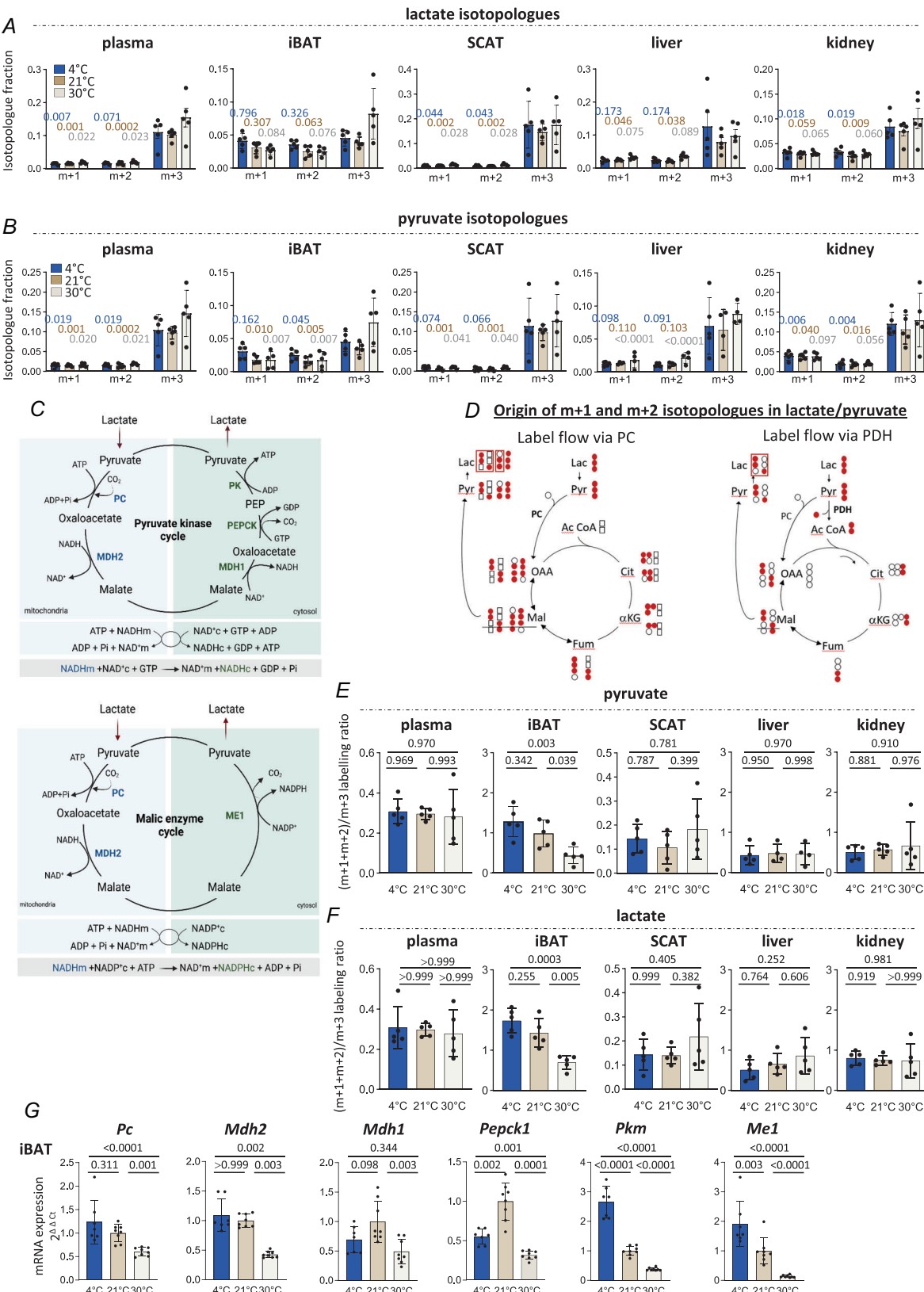

**Figure 6.  Thermoneutral housing decreases pyruvate cycling in brown adipose tissue**
*A*, fractions of m + 1, m + 2 and m + 3 isotopologues of lactate in plasma, iBAT, SCAT, liver and kidney after
intraperitoneal injection of [U-$^{13}$C]-lactate to 6 h fasted mice previously exposed to 4°C or housed at 21 or 30°C

($n$ = 5 per group). *Blue*: m + 3 *vs*. m + 1 and *vs*. m + 2 for mice exposed at 4°C; *brown*: m + 3 *vs*. m + 1 and *vs*. m + 2 for mice exposed at 21°C; *grey*: m + 3 *vs*. m + 1 and *vs*. m + 2 for mice exposed at 30°C. Plasma: *blue* by Kruskal–Wallis; *brown* by one-way ANOVA; *grey* by one-way ANOVA. iBAT: *blue* by one-way ANOVA; *brown* by one-way ANOVA; *grey* by one-way ANOVA. SCAT: *blue* by one-way ANOVA; *brown* by one-way ANOVA; *grey* by one-way ANOVA. Liver: *blue* by one-way ANOVA; *brown* by one-way ANOVA; *grey* by one-way ANOVA. Kidney: *blue* by one-way ANOVA; *brown* by Kruskal–Wallis; *grey* by one-way ANOVA. B, fractions of m + 1, m + 2 and m + 3 isotopologues of pyruvate in plasma, iBAT, SCAT, liver and kidney after intraperitoneal injection of [U-$^{13}$C]-lactate to 6 h fasted mice previously exposed to 4°C or housed at 21 or 30°C ($n$ = 4–5 per group). *Blue*: m + 3 *vs*. m + 1 and *vs*. m + 2 for mice exposed at 4°C; *brown*: m + 3 *vs*. m + 1 and *vs*. m + 2 for mice exposed at 21°C; *grey*: m + 3 *vs*. m + 1 and *vs*. m + 2 for mice exposed at 30°C. Plasma: *blue* by one-way ANOVA; *brown* by one-way ANOVA; *grey* by one-way ANOVA. iBAT: *blue* by one-way ANOVA; *brown* by one-way ANOVA; *grey* by one-way ANOVA. SCAT: *blue* by one-way ANOVA; *brown* by one-way ANOVA; *grey* by one-way ANOVA. Liver: *blue* by one-way ANOVA; *brown* by one-way ANOVA; *grey* by one-way ANOVA. Kidney: *blue* by one-way ANOVA; *brown* by one-way ANOVA; *grey* by one-way ANOVA. C, schematics illustrating pyruvate kinase and malic enzyme cycles. D, schematics illustrating the origin of m + 1 and m + 2 isotopologues in lactate and pyruvate. PDH, pyruvate dehydrogenase; PC, pyruvate carboxylase; Lac, lactate; Pyr, pyruvate; OAA, oxaloacetate; Ac-CoA, acetyl-CoA; Cit, citrate; aKG, alpha-ketoglutarate; Fum, fumarate; Mal, malate. E, value of the (m + 1 + m + 2)/m + 3 isotopologue ratio of pyruvate in iBAT, SCAT, liver and kidney after intraperitoneal injection of [U-$^{13}$C]-lactate to 6 h fasted mice previously exposed at 4°C or housed at 21 or 30°C ($n$ = 5 per group, one-way ANOVA). F, value of the (m + 1 + m + 2)/m + 3 isotopologue ratio of lactate in iBAT, SCAT, liver and kidney after intraperitoneal injection of [U-$^{13}$C]-lactate to 6 h fasted mice previously exposed at 4°C or housed at 21 or 30°C ($n$ = 5 per group, plasma: Kruskal–Wallis, iBAT, SCAT, liver, kidney: one-way ANOVA). G, mRNA levels of *Pc* (pyruvate carboxylase), *Mdh2* (malate dehydrogenase-2), *Mdh1* (malate dehydrogenase-1), *Pepck1* (phosphoenolpyruvate carboxykinase), *Pkm* (pyruvate kinase) and *Me1* (malic enzyme) in iBAT of 6 h fasted mice previously exposed at 4°C or housed at 21 or 30°C ($n$ = 8 per group, *Pc*, *Mdh1*, *Pepck1*, *Pkm*, *Me1*: one-way ANOVA; *Mdh2*: Kruskal–Wallis). Mice were 4-month-old males. Data are presented as mean ± SD.

mammalian cells. In both cases, there is utilization and re-synthesis of pyruvate, which can be evaluated from the (m + 1 + m + 2)/m + 3 labelling ratio, considered as a proxy of pyruvate cycling (the higher the ratio, the higher the recycling). Although no modification occurred in 4°C mice compared to 21°C mice, we observed a strong decrease in the pyruvate (m + 1 + m + 2)/m + 3 labelling ratio in iBAT at thermoneutrality and no changes in plasma, SCAT, liver and kidney (Fig. 6*E*). Similar profiles were found for the lactate (m + 1 + m + 2)/m + 3 labelling ratio, indicating that lactate cycling occurs as well (Fig. 6*F*).

Both pyruvate kinase and malic enzyme cycles (Fig. 6*C*) share some of the same enzymes [mitochondrial pyruvate carboxylase (PC) and malate dehydrogenase (MDH2)] and use specific ones [cytosolic MDH1, PEPCK1 and pyruvate kinase (PK) for the pyruvate kinase cycle, and malic enzyme (ME1) for the malic enzyme cycle]. In iBAT, expression of *Pc* and *Mdh2* did not differ between 4 and 21°C mice, but was significantly downregulated in 30°C mice (Fig. 6*G*), resembling the response observed for pyruvate cycling (Fig. 6*E*, 6*F*). With the exception of cytosolic *Mdh1* and *Pepck1*, *Pkm* and *Me1* expression levels were decreased in iBAT in 21°C mice relative to 4°C mice, and in 30°C mice relative to 21°C mice (Fig. 6*G*). This suggests that two cycles exist in iBAT although these cycles may be only partially regulated at the transcriptional level. Taken together, our findings demonstrate that a pyruvate metabolic cycling process is highly active in iBAT but repressed when organismal metabolic heat production and heat loss are balanced.

## Discussion

This study demonstrates that BAT activity impacts systemic lactate clearance. Lactate clearance in mice appears to be strongly regulated by thermogenic activity, increasing in mice experiencing prolonged cold exposure and decreasing as thermoneutral housing temperature is reached. Lactate clearance is also reduced in the absence of UCP1, although the phenotype of *Ucp1*$^{KO}$ mice – which are on a C57BL/6J background as originally generated by the group of Leslie Kozak (Enerback et al., 1997) – might result from both UCP1 deficiency and dysregulation of mitochondrial components (Kazak et al., 2017) and adaptive mechanisms described in this mouse model (Kazak et al., 2015; Keipert et al., 2015; Oeckl et al., 2022). Future studies using alternative *Ucp1*$^{KO}$ models on other genetic backgrounds [129S1/SvImJ or 129S1/SvImJ × C57BL/6J hybrid backgrounds that exhibit different phenotypes and cold sensitivity (Hofmann et al., 2001), the latter exhibiting similar downregulation of mitochondrial components to the *Ucp1*$^{KO}$ mice in the C57Bl6 background (Rahbani et al., 2024) or newly described inducible models (Rahbani et al., 2024)] would be valuable to expand upon our findings. The relationship between thermogenic state and lactate clearance suggests that lactate utilization at an organismal level varies based on metabolic demands associated with thermoregulation, and highlights the role of lactate in this process, as reported by others (Bornstein et al., 2023; Park et al., 2023). Lactate production from white adipose tissues was reported to impact on circulating lactate levels in humans (DiGirolamo et al., 1992) and in *Drosophila* (Krycer et al.,

2020). Our findings also suggest a significant – and much less appreciated – impact of thermogenic adipose tissues activity on systemic lactate clearance. Our study was dedicated to the characterization of lactate utilization in different tissues following a lactate load and across different thermogenic states. Whether BAT acts as a net lactate importer or exporter has been previously addressed in a few studies in rodents as well as in humans. Microdialysis experiments showed net lactate release by BAT in lean healthy men (Weir et al., 2018), in accordance with a previous study performed in rats (Ma & Foster, 1986) but not with a recent study performed in mice where net lactate import has been shown (Park et al., 2023). These discrepancies may reflect species but also experimental (fed/fasted state, duration and degree of cold exposure) as well as age- and sex-related putative differences, highlighting the need to better understand whether BAT acts as a net lactate importer or exporter across different physiological conditions, including in humans. The pathophysiological consequences might be broad given: (i) the pleiotropic roles of lactate (Brooks, 2018), (ii) the fact that lactate can contribute to muscle and liver pathogenesis (Choi et al., 2002; Rho et al., 2023) when not properly regulated and (iii) that circulating lactate has important impacts on tissue redox balance and homeostasis (Patgiri et al., 2020; Rabinowitz & Enerback, 2020).

According to the thermogenic state, in parallel with the regulation of brown and beige fat thermogenic activity, coordinated inter-organ communication takes place to sustain energy requirements for thermogenesis (Bornstein et al., 2023). The expression of lactate metabolism genes is very sensitive to thermogenic conditions in both brown and beige adipose tissues. These relationships suggest that they could directly contribute to the effect of housing temperature on blood lactate clearance. Despite the high expression of MCT1 in brown and beige adipocytes (Iwanaga et al., 2009; Lagarde et al., 2021), its deletion does not impact on lactate clearance, suggesting: (i) that lactate transport through MCT1 in adipose tissues is not sufficient to impact on circulating lactate levels, (ii) that additional transporters including other MCT isoforms or transporters from other families such as SLC5A12 (Pucino et al., 2019) may be involved or compensate for the reduction of MCT1 in adipose tissues, and/or (iii) that brown and beige adipocytes regulate blood lactate metabolism in an indirect manner through an inter-organ communication mechanism. Even if lactate was shown to be the second most important contributor – after glucose – to carbon influx in BAT (Park et al., 2023), it does not necessarily mean that BAT regulates blood lactataemia directly by consuming it massively from the blood. In support of this, we found that lactate contribution to gluconeogenesis increased after adaptation to long-term cold exposure that could explain the increased lactate clearance of 4°C mice. These data are in accordance with a recent study (Bornstein et al., 2023) that demonstrated an increased glucose generation from lactate upon acute cold exposure and the important role of gluconeogenesis in thermogenesis (Bornstein et al., 2023). Given that BAT uses large quantities of glucose when activated, any change in its activity could then regulate the ability of the liver and/or kidney to use lactate for gluconeogenesis, through released metabolic and/or endocrine messengers, mechanisms recently demonstrated as supporting BAT–liver inter-organ dialogue (Qing et al., 2020; Xu et al., 2023). Specific transgenic mouse models targeting actors of lactate metabolism in a tissue-specific manner should help to resolve the existence of such complex inter-organ metabolic communications.

Characterization of lactate metabolic fate during the clearance phase highlighted that, although chronic exposure to 4°C did not have any impact on the relative contribution of lactate to the TCA cycle in BAT compared to 21°C mice as demonstrated in mice acutely exposed to cold (Bornstein et al., 2023), it decreased significantly in BAT in thermoneutral conditions. One could postulate that lactate could be re-routed toward *de novo* lipogenesis in this condition, as lactate has been shown to fuel *de novo* lipogenesis (Chen et al., 2016) including in adipose tissues (Katz & Wals, 1974). Specific experiments enabling $^{13}C$ detection in lipids should enable this hypothesis to be tested. $^{13}C$-lactate tracer experiments also indicate that a robust pyruvate cycling process in BAT is fed by lactate, and is more active than in SCAT, liver and kidney. Pyruvate cycling requires first the activity of pyruvate carboxylase, an anaplerotic enzyme that serves to replenish the TCA cycle (Owen et al., 2002) and which plays an important role in BAT metabolic activity (Bornstein et al., 2023; Cannon & Nedergaard, 1979; Lao-On et al., 2021). Regeneration of cytosolic pyruvate could result from either the pyruvate kinase cycle or from the malic enzyme cycle (Fig. 6C), both cycles representing a redox transfer mechanism, and enabling the transfer of electrons from mitochondrial NADH to a cytosolic one ('pyruvate kinase cycle') or to NADPH ('malic enzyme cycle'). While cytosolic NADH could feed glyceroneogenesis, a process highly active in BAT (Reshef et al., 2003), NADPH could be used for redox homeostasis and lipid synthesis, through NADPH supply to glutathione peroxidase and fatty acid synthase activities respectively. While further studies will be needed to characterize the molecular mechanisms involved in pyruvate cycling, including through monitoring protein levels and activity of the putative targets and invalidating them using tissue-specific transgenic mice models, a recent report identified the existence of a so-called hydride transfer complex, composed of pyruvate carboxylase, malate dehydrogenase and malic enzyme, which plays a key role in antioxidant defence by converting the redox power of NADH into NADPH, thereby promoting

the escape of cancer cells from senescence (Igelmann et al., 2021; Maus & Serrano, 2021). As the activation of BAT is associated with the generation of mitochondrial reactive oxygen species (Chouchani et al., 2016), the pyruvate cycling process fed by lactate identified here could support redox homeostasis in BAT. Because this cycle consumes one ATP/GTP, it could be considered, as for calcium, creatine and lipid cycling mechanisms (Ikeda et al., 2017; Kazak et al., 2015; Mottillo et al., 2014; Roesler & Kazak, 2020; Sharma et al., 2024), as another metabolic thermogenic cycle important for BAT biology.

Thermoneutral housing has drastic consequences on whole-body physiology in mice and notably on brown and beige adipose tissue biology (Ganeshan & Chawla, 2017; James et al., 2023). Indeed, housing mice in thermoneutral conditions reduces sympathetic-nerve-derived noradrenaline (Cui et al., 2016) and induces macrophage accumulation in brown fat – although this inflammation does not impair subsequent cold-induced activation (Fischer et al., 2020). The impact of housing temperature in terms of translational relevance is supported by related studies, such as how the benefits of exercise training in mice depend on housing temperature, and that exercise-induced increased glucose tolerance and beiging of white adipose tissues are partly lost when mice were housed in thermoneutral conditions (Raun et al., 2020).

This study has some limitations. We aimed to characterize lactate metabolic fate in tissues during the clearance phase, in exactly the same conditions than for the lactate tolerance test and thus performed bolus injection of $^{13}$C-lactate. Although a long-term perfusion of a low concentration of lactate is required to calculate metabolite fluxes without altering blood lactate concentration (Hui et al., 2017, 2020), bolus injection enables researchers to follow the fate of a labelled metabolite rapidly after its injection and determine the relative contribution of its carbons to downstream metabolites, that is, to investigate the short-term metabolic fate of nutrients. This method was previously applied to decipher the metabolic fate of glucose or succinate in BAT (Jung et al., 2021; Mills et al., 2018; Reddy et al., 2024; Wang, Ning et al., 2020) or lactate in the liver (with very similar experimental conditions compared to our study; Roichman et al., 2021). However, the change in blood lactate concentration after bolus injection may have a significant impact on tissue metabolic activity including through osmolarity changes (Lund et al., 2023). Future studies should aim to better understand the mechanisms behind systemic lactate clearance, notably through deeper proteomic and enzymatic profiling and invalidation of specific enzymes or transporters using tissue-specific transgenic mice models. All experiments performed in the present work have been done in male mice and whether sex affects lactate metabolism should be investigated in future studies.

Taken together, our study demonstrates that housing mice at thermoneutrality rewires systemic and brown fat lactate metabolism, highlighting the importance of considering housing temperature conditions when studying systemic lactate metabolism and inter-organ communication.

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

## Additional information

### Data availability statement

Data will be made available upon request to the lead contact.

### Competing interests

The authors declare that they have no known competing financial interests or personal relationships that could have appeared to influence the work reported in this paper.

## Author contributions

R.M., Y.J., D.L., S.K., L.P., A.K.B.S., L.C., J.C.P., I.A. and A.C. conceived and designed the study. R.M., Y.J., D.L., S.K., L.P.B., J.N., M.S., M.P., E.H., R.A.D.S. and A.C. performed experiments. R.M., Y.J., D.L., S.K., L.P.B., M.S., M.P., I.R.L., E.H., R.A.D.S., J.C.P., I.A. and A.C. analysed the data. R.M., Y.J., D.L., S.K., J.N., M.P., I.R.L., E.H., R.A.D.S., A.G., L.P., A.K.B.S., J.P.P., C.M., L.C., A.Y., C.D., J.C.P., I.A. and A.C. interpreted the results of the experiments. All authors wrote, edited, revised and approved the final version of the manuscript.

## Funding

This research was funded, in whole or in part, by the French National Research Agency (ANR), projects ANR-18-CE18-0006 and ANR-23-CE52-0008-01. This work was performed in the context of the INSPIRE Program [grants Région Occitanie Pyrénées-Méditerranée, reference number: 1 901 175, and European Regional Development Fund (ERDF), Project number: MP0022856] and of the IHU HealthAge of Toulouse [French National Research Agency (ANR) as part of the France 2030 programme (reference number: ANR 23-IAHU-0011)]. Rémi Montané and Damien Lagarde obtained PhD fellowhips from the French Ministry for Higher Education, Research and Innovation. This study was partially supported through the grant EUR CARe No. ANR-18-EURE-0003 in the framework of the Programme des Investissements d'Avenir. Raphael Alves de Souza obtained a PhD fellowhip from the CARe – Graduate School. LabHPEC and Melissa Parny were supported by the ANR project INBS (Infrastructure Nationale en Biologie Santé) ECellFrance (PIA – ANR-11-INBS-005). Jean Nakhle obtained a postdoctoral fellowship from INCa (PLBIO 2020-010, DIALAML; J. E. Sarry). MetaboHub-MetaToul (Metabolomics & Fluxomics facilities, Toulouse, France; http://www.metatoul.fr) is part of the French national infrastructure MetaboHUB and is funded by the ANR with grant number MetaboHUB-ANR-11-INBS-0010. Anne-Karine Bouzier-Sore and Luc Pellerin have received financial support from ANR project ANR-21-CE44-0023. Anne-Karine Bouzier-Sore's work was conducted in the framework of the University of Bordeaux's France 2030 programme RRI 'IMPACT' that received financial support from the French government. Cedric Moro received funding from EFSD/Boehringer Ingelheim European Research Programme on 'Multi-System Challenges in Diabetes' 2023, Agence Nationale de la Recherche (ANR-21-CE14-0057-01) and Fondation pour la Recherche Médicale (EQU202303016316).

## Acknowledgements

We thank all members of mice core facilities (UMS006, ANEXPLO, Inserm, Toulouse) in particular Marie Lulka and Cédric Baudelin for their support and technical assistance. Imaging and molecular analyses were performed on equipment from the 'Centre d'Expertise et de Ressources Technologiques' (CERT) from RESTORE UMR 1301-INSERM 5070-CNRS, and we thank in particular Emmanuelle Arnaud for molecular studies and Corinne Barreau for help with confocal imaging. We acknowledge the TRI-RESTORE (Genotoul-TRI), member of the national infrastructure France-BioImaging supported by the French National Research Agency (ANR-24-INBS-0005 FBI BIOGEN). We thank Christophe Guissard, Livia Robert and Garance Castino for help with experimentations. We also thank Emmanuel Gras (LHFA, Toulouse) and Arnaud Mourier (IBGC-CNRS, Bordeaux) for helpful discussions. Lara Gales and Floriant Bellvert (MetaToul, Toulouse, France) are gratefully acknowledged for fruitful discussions. We thank Stephan Offersman and Leslie Kozak for sharing the adiponectin CRE[ERT2] and *Ucp1*[KO] mouse models, respectively. We thank Emmanuelle Tena, Zoely Rakotomanga-Rajaonah, Gwenaelle Guérin and Suzy Bignau for their daily help with the administrative and financial management of our team. We thank the University of Lausanne for granting the permission to use MCT1 floxed mice. We are grateful to Life Sciences Editors for editing services.

## Keywords

brown adipose tissue, cold exposure, isotopic tracing experiments, lactate metabolism, pyruvate cycling, thermo-neutrality

## Supporting information

Additional supporting information can be found online in the Supporting Information section at the end of the HTML view of the article. Supporting information files available:

**Peer Review History**
**Supporting Information**

