## [Peer Review History · The Journal of Physiology]

Brown adipose tissue activity impacts systemic lactate clearance in male mice

Rémi Montané, Yannick Jeanson, Damien Lagarde, Spiro KHOURY, Léana Porcher-Bibes, Jean Nakhle, Marie Sallese, Mélissa Parny, Isabelle Raymond-Letron, Emma Huard, Raphael Alves de Souza, Anne Galinier, Luc Pellerin, Anne-Karine Bouzier Sore, Jean-Philippe Pradère, Cedric Moro, Louis Casteilla, Armelle Yart, Cedric Dray, Jean-Charles Portais, Isabelle Ader, and Audrey Carriere

DOI: 10.1113/JP288871

Corresponding author(s): Audrey Carriere (audrey.carriere-pazat@inserm.fr)

Review Timeline:

Submission Date:	14-Mar-2025
Editorial Decision:	09-Apr-2025
Revision Received:	15-Jul-2025
Editorial Decision:	19-Aug-2025
Revision Received:	20-Aug-2025
Accepted:	02-Sep-2025

Senior Editor: Paul Greenhaff

Reviewing Editor: Max Petersen

Transaction Report:

Dear Dr Carriere,

Re: JP-RP-2025-288871 "**Brown adipose tissue activity impacts systemic lactate clearance**" by Rémi Montané, Yannick Jeanson, Damien Lagarde, Spiro Khoury, Léana Porcher-Bibes, Jean Nakhle, Marie Sallèse, Mélissa Parny, Isabelle Raymond-Letron, Emma Huard, Raphael Alves de Souza, Anne Galinier, Luc Pellerin, Anne-Karine Bouziers Sore, Jean-Philippe Pradère, Cedric Moro, Louis Casteilla, Armelle Yart, Cedric Dray, Jean-Charles Portais, Isabelle Ader, and Audrey Carriere

Thank you for submitting your manuscript to The Journal of Physiology. It has been assessed by a Reviewing Editor and by 2 expert referees and we are pleased to tell you that it is potentially acceptable for publication following satisfactory major revision.

REVISION CHECKLIST:

Upload a full Response to Referees file. To create your 'Response to Referees': copy all the reports, including any comments from the Senior and Reviewing Editors, into a Microsoft Word, or similar, file and respond to each point, using

font or background colour to distinguish comments and responses and upload as the required file type.

We look forward to receiving your revised submission.

Yours sincerely,

Paul Greenhaff
Senior Editor
The Journal of Physiology

REQUIRED ITEMS

- Author photo and profile. First or joint first authors are asked to provide a short biography (no more than 100 words for one author or 150 words in total for joint first authors) and a portrait photograph. These should be uploaded and clearly labelled together in a Word document with the revised version of the manuscript. See Information for Authors for further details.

- The contact information for the person responsible for 'Research Governance' at your institution needs to be provided. This includes their name and an institutional email address. Please ensure the contact is not an author on this paper and provide an alternate contact if necessary, or confirm in the submission form that the author whose email was provided has sole responsibility for research governance. This is the person who is responsible for regulations, principles and standards of good practice in research carried out at the institution, for instance the ethical treatment of animals, the keeping of proper experimental records or the reporting of results.

- You must start the Methods section with a paragraph headed Ethical approval (https://jp.msubmit.net/cgi-bin/main.plex?form_type=display_requirements#methods).

Research must comply with The Journal's policies regarding animal experiments (<https://physoc.onlinelibrary.wiley.com/hub/animal-experiments>) and adherence to these policies must be stated in the manuscript.

Authors should confirm in their Methods section that their experiments were carried out according to the guidelines laid down by their institution's animal welfare committee, including an ethics approval reference number. The Methods section must contain a statement about access to food, water and housing, details of the anaesthetic regime: anaesthetic used, dose and route of administration, and method of killing the experimental animals.

- Please upload separate high-quality figure files via the submission form.

- You must upload original, uncropped western blot/gel images (including controls) if they are not included in the manuscript. This is to confirm that no inappropriate, unethical or misleading image manipulation has occurred. These should be uploaded as 'Supporting information for review process only'. Please label/highlight the original gels so that we can clearly see which sections/lanes have been used in the manuscript figures. For more information, see: <https://physoc.onlinelibrary.wiley.com/hub/journal-policies#imagmanip>.

- Please ensure that any tables are editable and in Word format, and wherever possible, embedded in the article file itself.

- Please ensure that the Article File you upload is a Word file.

- Your paper contains Supporting Information of a type that we no longer publish, including supplementary tables and figures. Any information essential to an understanding of the paper must be included as part of the main manuscript and figures. The only Supporting Information that we publish are video and audio, 3D structures, program codes and large data files. Your revised paper will be returned to you if it does not adhere to our Supporting Information Guidelines.

- Papers must comply with the Statistics Policy: https://jp.msubmit.net/cgi-bin/main.plex?form_type=display_requirements#statistics.

In summary:

- If n {less than or equal to} 30, all data points must be plotted in the figure in a way that reveals their range and distribution. A bar graph with data points overlaid, a box and whisker plot or a violin plot (preferably with data points included) are acceptable formats.

- If $n > 30$, then the entire raw dataset must be made available either as supporting information, or hosted on a not-for-profit repository, e.g. FigShare, with access details provided in the manuscript.

- 'n' clearly defined (e.g. x cells from y slices in z animals) in the Methods. Authors should be mindful of pseudoreplication.

- All relevant 'n' values must be clearly stated in the main text, figures and tables.

- The most appropriate summary statistic (e.g. mean or median and standard deviation) must be used. Standard Error of the Mean (SEM) alone is not permitted.

- Exact p values must be stated. Authors must not use 'greater than' or 'less than'. Exact p values must be stated to three significant figures even when 'no statistical significance' is claimed.

- Please include an Abstract Figure file, as well as the Figure Legend text within the main article file. The Abstract Figure is a piece of artwork designed to give readers an immediate understanding of the research and should summarise the main conclusions. If possible, the image should be easily 'readable' from left to right or top to bottom. It should show the physiological relevance of the manuscript so readers can assess the importance and content of its findings. Abstract Figures should not merely recapitulate other figures in the manuscript. Please try to keep the diagram as simple as possible and without superfluous information that may distract from the main conclusion(s). Abstract Figures must be provided by authors no later than the revised manuscript stage and should be uploaded as a separate file during online submission labelled as File Type 'Abstract Figure'. Please also ensure that you include the figure legend in the main article file. All Abstract Figures should be created using BioRender. Authors should use The Journal's premium BioRender account to export high-resolution images. Details on how to use and access the premium account are included as part of this email.

EDITOR COMMENTS

The referees were both enthusiastic about the potential of the manuscript but identified distinct and important areas to address in revision.

Senior Editor:

Thank you for the manuscript submission to The Journal of Physiology. It has been considered by a reviewing editor and two expert reviewers. Both reviewers are enthusiastic about the potential of the manuscript and believe it could be impactful. However, both have identified a number of points that will need to be addressed by the authors, including the inclusion of additional experimental findings. This will increase the impact of the research and hopefully the authors are positive about such additions.

From a housekeeping perspective please be clearer about the methods of euthanasia so as to comply with The Journal of Physiology guidelines. Additionally, please note the statistics policy - data are currently shown as mean \pm SEM and exact P values are not provided.

REFEREE COMMENTS

Referee #1:

Montane and colleagues present the report investigating the role of brown and beige adipose tissue on lactate clearance and utilisation. In this work they have undertaken i.p. lactate tolerance tests to mice housed at either 4, 21 or 30°C. They found that cold exposure increased systemic lactate clearance in fasted and fed mice (in parallel with increased *Ldha/b* and *Mct1* expression), and that clearance was reduced in UCP1 KO mice. They also found that lactate tolerance tests did not reveal any difference in mice with adipose tissue specific knockout of the lactate transporter MCT1. Finally, they have undertaken repeated lactate tolerance tests with labelled lactate to determine the metabolic fate in BAT/WAT. They show that there is some reduction in labelled TCA cycle metabolites and also m+1 lactate in the 30°C housed animals, in keeping with reduced LDH. There is significant interest in the role of lactate in whole body metabolism and in thermogenesis, the studies are well designed including the use of an iso-osmolar control, the paper well written and the authors highlight some of the limitations such as the lack of steady state tracer infusions. I have the following specific points for the authors to address.

MAJOR COMMENTS

- 1) The authors have knocked out *Mct1* and found no difference in lactate clearance, and conclude that MCT1 in WAT/BAT does not contribute to systemic lactate clearance. However, they speculate in the discussion that there could be increased uptake by other MCT transporters as a compensatory mechanism which seems highly plausible. They have measured mRNA levels of these transporters in the first cohort showing increased expression in the cold exposed mice, and it would strengthen the paper to measure these genes in the AT depots of the MCT1 WT and KO mice please to support this conclusion.
- 2) The unlabelled lactate tolerance tests increase circulating lactate levels ~4-fold so are responsible for the majority of the circulating lactate, yet the authors state in the labelled lactate tolerance test that only 10% of circulating lactate was labelled, despite apparently using identical concentrations which seems surprising, when you may have expected the majority to be labelled. How do the authors explain this please? It is also interesting that the lactate enrichment in the BAT and WAT tissues are identical between 30 and 4°C groups 15 minutes after injection when tissues were collected, suggesting that lactate uptake by BAT may be unchanged by room temperature. Could this mean that other tissues are mediating the increased systemic lactate clearance in the 4 and 21°C groups? Did the authors collect other tissues to look at uptake/ utilisation by other key metabolic tissues such as liver, this would strengthen the paper? This is important in particular as the adipose MCT1 KO doesn't show reduced lactate clearance systemically. Of course there may be rapid utilisation of lactate by the tissue as the authors already note substantial cycling through the TCA cycle, but it is interesting that BAT may not drive this clearance and the role of other tissues should be discussed in more depth please.
- 3) The authors note less cycling through the TCA cycle in the 30°C mice as to be expected when thermogenesis is not required, but there is substantial uptake of lactate by BAT and potentially similar uptake. Did the authors look for incorporation into other pathways than the TCA cycle as has been seen for e.g. glucose (PMID 34320357), it would be interesting to determine what lactate is used for at thermoneutrality, or at least discuss the possible fates of lactate in more depth please.
- 4) The authors are purely exploring the uptake of lactate during different ambient conditions in their model, but do reference previous work stating that lactate can be released by BAT during thermogenesis (Ma and Foster, 1986). BAT has also been reported to be a net exporter of lactate during thermoneutral conditions and thermogenesis in humans (PMID 29805098), so it would be important for the authors to discuss the translation of their findings to humans please, including the possible role of lactate generation and release by BAT in the different ambient conditions.
- 5) It's interesting that the lactate concentrations in the UCP1 KO mice housed at 21°C by 30 minutes are substantially higher than the levels in the wild type mice housed at even 30°C, in keeping with markedly reduced clearance. Can the authors explain this observation please? It could have been interesting to house the UCP1 KOs at 30°C to see whether this reduced clearance persisted vs WT.

MINOR COMMENTS

- 1) Title - It would be important to add 'in male mice' to the title to make it clear to the reader the model studied please.
- 2) In figure 4D, it would be helpful for the reader if the authors could write MCT1^{WT} and MCT1^{ΔAd} on the respective images please.

Referee #2:

In this study, Montané et al. investigated the impact of BAT stimulation on lactate uptake and intra-tissue metabolism. Male C57Bl/6J mice were housed at 21{degree sign}C, exposed to 4{degree sign}C during 7 days or housed at 30{degree sign}C for 2 months, then underwent i.p. lactate tolerance tests, without or with 13C-labeling, under fasted (6h) or fed conditions to perform metabolic isotope tracing experiments. The authors report that animals housed at 30{degree sign}C, whereby BAT is considered largely inactive and show reduced Ucp1 expression, present with reductions in lactate clearance to a similar extent as Ucp1ko mice, while those housed at 4{degree sign}C showed improved clearance compared to mice housed at 21{degree sign}C. Housing temperature appeared to impact the gene expression of certain genes associated with lactate metabolism or proton-linked transport of metabolites like lactate or pyruvate, although Mct1ko mice did not present with any changes in lactate clearance. Finally, based on the metabolic isotope tracing experiments, the authors suggest that the enrichment pattern whereby there is a presence of m+1 and m+2 lactate species implies a cycling between pyruvate and lactate, which appears to be greater in mice housed at 21{degree sign}C and 4{degree sign}C than when housed at 30{degree sign}C.

Critique

This study provides important observational insights into the uptake and possible metabolic fate of lactate, a seemingly important metabolite and substrate for BAT thermogenesis. There is mounting evidence demonstrating the role of lactate as a signaling molecule as well as a critical metabolic substrate, but little is known about how it is used once taken up by thermogenic organs like BAT. In this regard, this manuscript provides an important starting point to begin exploring the latter. However, there are important methodological and interpretational limitations that should be considered and possibly addressed.

(1) Both the unlabeled and labeled lactate were administered by i.p. - this likely has a significant impact in the metabolism of lactate compared to i.v. administration. What was the justification for this administration and how does its metabolism differ from i.v.? There are apparent differences in lactate appearance between i.p. vs. s.c. administration, which is also sex-dependent (PMID: 31934509), is this also the case for i.p. vs. i.v. administration?

(2) The interpretation of the metabolic isotope tracing experiments are more complex than how it is presented. First, currently, there is a lack of evidence supporting the claims and an absence of confirmatory experiments to validate them. As it stands, it remains largely observational and open to many interpretations. The authors could help with interpretation by inhibiting pathways or silencing certain enzymes or transporters. Aside from using Mct1 knockout mice, this is the extent of probing that was done and these mice likely display some compensatory mechanisms.

Second, there are critical products of lactate or pyruvate metabolism that are missing, such as acetyl CoA or citrate, carbon dioxide, malate, fumarate, aspartate (though it is in the supplemental files) and PEP. If referring to pyruvate carboxylation, then minimally, oxaloacetate should be presented as should malate and PEP.

Third, gene expression tells us very little about the metabolism of an organ. There are many regulatory steps between gene expression and protein/enzymatic function. Minimally, we need to see protein levels and functional outcomes.

(3) Reporting of data

- Given that 13C-lactate will be rapidly metabolized, the data reported and normalized to tissue 13C-lactate will be highly influenced by a denominator effect, making it very difficult to interpret. Why not present the fractional enrichments (as is done in Table S1), or normalize to the injected dose or relative to total 13C?
- There is an insufficient sample size to complete the statistics presented.
- Given the dynamics of 13C-lactate, it would be important to look at both the mitochondrial and cytosolic pools of metabolites, not just the tissue as a whole, to make sense of what might be happening.
- Standard deviation should be reported rather than SEM.

(4) Methods

- As currently written, the Methods lack some important details - how were animals euthanized? Was lactate tolerance performed under anaesthesia or awake? If so, what anaesthetic was used, how long, what dose? Presumably, the lactate tolerance tests were performed at ambient temperature, rather than their housed temperatures? How could this impact the outcomes?
- Are the Ucp1 ko mice on the C57 or 129/sv or hybrid background? The original Ucp1 ko mice have serious mitochondrial defects (see PMID: 28630339). Although cited and referenced in the Discussion, it is unclear which model is ultimately used and why it would be selected over more appropriate Ucp1 ko models currently available.
- Pg. 12, 1st paragraph - the authors indicate that "Because 13C lactate was the only 13C source, the relative contribution

from the tracer to downstream metabolites was calculated by dividing the enrichment of the metabolite with the enrichment of the tracer (Wang et al., 2020b)." Is this in reference to enrichment of tracer in plasma, in tissue or of infusate?

- If animals were group housed, ambient temperature (21C) may not be that cold.
- It is still unclear how pyruvate cycling is calculated? The cycle and how it is determined needs to be described more completely. Simply an $(m+1 + m+2)/m+3$ calculation is not a reflection of pyruvate cycling.

Specific comments:

(1) Pg 4, ln 1 - thermogenesis and heat production are synonymous, so thermogenesis can't trigger heat production.

(2) Pg 4, 1st paragraph - BAT doesn't become inactive when heat production and heat loss are balanced. It becomes inactive when the sympathetic stimulation is reduced, which would happen when stimulus (heat loss) is removed.

(3) Pg 4, 2nd paragraph - UCP1 uncouples cellular respiration from ATP synthesis.

(4) Pg 4, 2nd paragraph - beige adipocytes don't produce the same amount of heat as brown adipocytes and seem to also rely on other futile cycles like calcium cycling and creatine cycling.

(5) Pg. 10 - Ldha and Ldhd mRNA are not enzymes and therefore cannot be involved in the reversible conversion of pyruvate into lactate.

(6) Pg 12 - 2nd paragraph - authors state that "Because ^{13}C lactate was the only ^{13}C source, the relative contribution from the tracer to downstream metabolites was calculated by dividing the enrichment of the metabolite with the enrichment of the tracer (Wang et al., 2020b)." However, lactate is rapidly metabolized, so why would we assume that 15 min post injection under possibly stimulated states, we would mainly expect lactate isotopologue to be $m+3$ lactate?

(7) Pg 12 - 2nd paragraph - couldn't a singly and doubly labeled species reflect a loss of carbon to CO_2 ?

END OF COMMENTS

REFeree COMMENTS

Referee #1:

Montane and colleagues present the report investigating the role of brown and beige adipose tissue on lactate clearance and utilisation. In this work they have undertaken i.p. lactate tolerance tests to mice housed at either 4, 21 or 30{degree sign}C. They found that cold exposure increased systemic lactate clearance in fasted and fed mice (in parallel with increased *Ldha/b* and *Mct1* expression), and that clearance was reduced in UCP1 KO mice. They also found that lactate tolerance tests did not reveal any difference in mice with adipose tissue specific knockout of the lactate transporter MCT1. Finally, they have undertaken repeated lactate tolerance tests with labelled lactate to determine the metabolic fate in BAT/ WAT. They show that there is some reduction in labelled TCA cycle metabolites and also m+1 lactate in the 30{degree sign}C housed animals, in keeping with reduced LDH. There is significant interest in the role of lactate in whole body metabolism and in thermogenesis, the studies are well designed including the use of an iso-osmolar control, the paper well written and the authors highlight some of the limitations such as the lack of steady state tracer infusions. I have the following specific points for the authors to address.

Reply. We thank the Reviewer for their positive assessment of our study.

MAJOR COMMENTS

1) The authors have knocked out *Mct1* and found no difference in lactate clearance, and conclude that MCT1 in WAT/BAT does not contribute to systemic lactate clearance. However, they speculate in the discussion that there could be increased uptake by other MCT transporters as a compensatory mechanism which seems highly plausible. They have measured mRNA levels of these transporters in the first cohort showing increased expression in the cold exposed mice, and it would strengthen the paper to measure these genes in the AT depots of the MCT1 WT and KO mice please to support this conclusion.

Reply. As suggested by the Reviewer, we measured gene expression of *Mct2* and *Mct4* isoforms in adipose tissues of adipose tissue-specific *Mct1* knock out mice. We found similar mRNA levels of both *Mct2* and *Mct4* in iBAT and SCAT of *Mct1*^{ΔAd} mice compared to *Mct1*^{WT} mice, suggesting no compensatory impact of the loss of MCT1 on the gene expression of these transporters. However, even if no impact of gene expression was observed, this does not exclude the possibility that these transporters or other transporters (from the MCT family or from others families) present in adipose tissues could compensate for the loss of MCT1. We included these data in the **new Figures 4D, 4E, 4F, 4G** of the manuscript and completed the Results section accordingly (**please see page 12 of the revised manuscript with track changes**).

2) The unlabelled lactate tolerance tests increase circulating lactate levels ~4-fold so are responsible for the majority of the circulating lactate, yet the authors state in the labelled lactate tolerance test that only 10% of circulating lactate was labelled, despite apparently using identical concentrations which seems surprising, when you may have expected the majority to be labelled. How do the authors explain this please?

Reply. It is indeed important to better explain this point. The measure of ¹³C lactate plasma enrichment has been done 15 minutes after the injection. The level of ¹³C labeling therefore represents the balance between the injected ¹³C lactate solution (at t=0) and endogenous ¹²C lactate production by all tissues of the organism (during 15 minutes from the time of the injection). This endogenous lactate production induced a dilution of the injected ¹³C lactate, resulting in 14% in average of ¹³C enrichment in plasma lactate. This dynamic reflects the rapid turnover and high endogenous production of lactate. Indeed, lactate is the metabolite that has the highest turnover as shown by Hui and colleagues (Hui S et al. 2017).

PMID: 29045397). This clarification has been added in the Results section (**please see page 13 of the revised manuscript with track changes**).

It is also interesting that the lactate enrichment in the BAT and WAT tissues are identical between 30 and 4°C groups 15 minutes after injection when tissues were collected, suggesting that lactate uptake by BAT may be unchanged by room temperature. Could this mean that other tissues are mediating the increased systemic lactate clearance in the 4 and 21°C groups? Did the authors collect other tissues to look at uptake/ utilisation by other key metabolic tissues such as liver, this would strengthen the paper? This is important in particular as the adipose MCT1 KO doesn't show reduced lactate clearance systemically. Of course there may be rapid utilisation of lactate by the tissue as the authors already note substantial cycling through the TCA cycle, but it is interesting that BAT may not drive this clearance and the role of other tissues should be discussed in more depth please.

Reply. We totally agree with the Reviewer with the fact that systemic lactate clearance may result from modification of lactate utilization in several organs and not only in BAT. As suggested, we included ¹³C labeling data from other key metabolic tissues such as liver and kidney. Although no difference was observed for the ¹³C labeling rate of lactate nor for the contribution of lactate to pyruvate and TCA related metabolites, we found a higher contribution of lactate to glucose in the liver of 4°C mice compared to 21°C and 30°C mice, concomitant with a significant increase in plasma ¹³C glucose enrichment in 4°C mice. A similar profile was observed in the kidney. Relative to standard housing conditions, thermoneutral housing did not have any impact on lactate contribution to gluconeogenesis. These findings are consistent with the higher mRNA levels of the cytosolic gluconeogenic phosphoenolpyruvate kinase enzyme (*Pepck1*), in both the liver and kidney of 4°C mice only. Together, these data indicate that lactate is a key contributor for fueling gluconeogenesis in mice adapted to prolonged cold exposure, a process that may contribute to the increased systemic lactate clearance of 4°C mice. These data are in accordance with a recent study (Bornstein et al., 2023, PMID: 37802078) that demonstrated an increased glucose generation from lactate upon acute cold exposure and the important role of gluconeogenesis in thermogenesis (Bornstein et al., 2023, PMID: 37802078). However, as gluconeogenesis fed by lactate was not decreased in 30°C mice compared to 21°C mice, a gluconeogenesis-independent mechanism may contribute to altering lactate clearance in the thermoneutral state. These findings have been included in the new **Figures 5A, 5B, 5E, 5F, 5G, 5H**, in the Results section (**please see pages 13/14 of the revised manuscript with track changes**) and in the Discussion (**please see page 16 of the revised manuscript with track changes**). The abstract and the end of the introduction (**please see page 6 of the revised manuscript with track changes**) were revised accordingly and a key point referring to these findings was included (**please see page 2 of the revised manuscript with track changes**).

Given the addition of these new data, we have generated a **new Figure 6** dedicated to pyruvate cycling. Data regarding pyruvate cycling that were previously shown in Figure 5 have been moved into this new Figure 6 (that has been further completed with new data in plasma, liver and kidney and pyruvate isotopologues distribution). These findings further highlight the specific profile of pyruvate cycling in BAT. These data are included in the new **Figures 6A, 6B, 6E, 6F** and in the Results section (**please see pages 14/15 of the revised manuscript with track changes**).

3) The authors note less cycling through the TCA cycle in the 30°C mice as to be expected when thermogenesis is not required, but there is substantial uptake of lactate by BAT and potentially similar uptake. Did the authors look for incorporation into other pathways than the TCA cycle as has been seen for e.g. glucose (PMID 34320357), it would be interesting to determine what lactate is used for at thermoneutrality, or at least discuss the possible fates of lactate in more depth please.

Reply. It would be indeed very interesting to decipher the metabolic fate of lactate in thermoneutral conditions in BAT. To provide a more complete overview on metabolic pathways fed by lactate, we analyzed ¹³C enrichment in six additional metabolites that could be detected in BAT. While no modification of ¹³C labeling enrichment in acetoacetate, trans-aconitate and 2 hydroxyglutarate was

observed between the different housing temperatures, a decreased ^{13}C enrichment in glycerol-3-phosphate, alanine and propionate was found in BAT of 30°C mice compared to 4°C mice (please see Figure 1 for reviewer).

Figure 1 for reviewers

These data further highlight that thermoneutral housing reduced lactate metabolism into several metabolic pathways in BAT, although not indicating its preferred metabolic fate in this condition. Our main hypothesis is that lactate contribution to *de novo* lipogenesis could be increased at thermoneutrality. It has been indeed shown that lactate can fuel lipid formation (Chen et al. 2016, PMID: 27618187) including in adipose tissues (Katz et Wals 1974, PMID: 4847561). We previously performed preliminary isotope-ratio mass spectrometry experiments but could not detect significant ^{13}C labeling in lipids, that could be due to the high amount of lipids in adipose tissues and therefore high dilution of the ^{13}C labeling. To go further on this, longer incubation times with ^{13}C lactate through perfusion experiments should be performed to favor ^{13}C enrichment in lipids therefore facilitating its detection, experiments that require very specific expertise, heavy adjustments and a dedicated study. We discussed the possible fate of lactate towards lipids in the Discussion section (please see page 17 of the revised manuscript with track changes).

4) The authors are purely exploring the uptake of lactate during different ambient conditions in their model, but do reference previous work stating that lactate can be released by BAT during thermogenesis (Ma and Foster, 1986). BAT has also been reported to be a net exporter of lactate during thermoneutral conditions and thermogenesis in humans (PMID 29805098), so it would be important for the authors to discuss the translation of their findings to humans please, including the possible role of lactate generation and release by BAT in the different ambient conditions.

Reply. As mentioned by the Reviewer, the present study was dedicated to the characterization of lactate utilization in different tissues following a lactate load and across different thermogenic states. Whether BAT acts as a net lactate importer or exporter has been previously addressed in a few studies in rodents as well as in human. Microdialysis experiments showed net lactate release by BAT in lean healthy men (Weir et al., 2018, PMID 29805098), in accordance with a previous study performed in rats (Ma & Foster, 1986, PMID: 3730946) but not with a recent work performed in mice where net lactate import has been shown (Park et al., 2023, PMID: 37337122). These discrepancies may reflect species but also experimental (fed/fasted state, duration and degree of cold exposure) as well as age and sex-related putative differences, highlighting the need to better understand whether BAT acts as a net lactate importer or exporter across different physiological conditions, including in humans. It is plausible that the direction of net lactate flux across BAT is highly context-dependent, influenced by both internal metabolic states and external environmental variations. Given the role of lactate in energy and redox homeostasis, any modification of its circulating levels depending on BAT activity and across different thermogenic states may have broad and important physiological consequences. Future studies are needed to resolve this. This comment has been added in the Discussion section (please see page 16 of the revised manuscript with track changes).

5) It's interesting that the lactate concentrations in the UCP1 KO mice housed at 21°C by 30 minutes are substantially higher than the levels in the wild type mice housed at even 30°C, in keeping with markedly reduced clearance. Can the authors explain this observation please? It could have been interesting to house the UCP1 KOs at 30°C to see whether this reduced clearance persisted vs WT.

Reply. As pointed by the Reviewer, *Ucp1*^{KO} mice exhibit higher blood lactate levels specially at the latter time points of the lactate tolerance test, compared to wild type mice housed at 30°C. Although both models (*Ucp1* genetic deficiency and thermoneutral housing) exhibit reduced lactate clearance, this highlight the fact that they are not completely equivalent. As mentioned in the Discussion, *Ucp1*^{KO} mice exhibit several adaptive mechanisms and phenotypes that might not be induced by thermoneutral housing of wild type animals. Furthermore, thermoneutral housing could impact other organs than BAT that may impact systemic lactate clearance, and this might be not reproduced in the case of *Ucp1* genetic deficiency.

As suggested by the Reviewer, we tested the impact of thermoneutral housing on systemic lactate clearance of *Ucp1*^{KO} mice. Data showed that *Ucp1*^{KO} mice exhibit very similar profile at both 21°C and 30°C, with the presence of a similar plateau at the later time points of the kinetic of the lactate tolerance test. Because thermoneutral housing did not further increase lactate intolerance of *Ucp1*^{KO} mice, this may suggest that both models share common mechanism(s) that contribute to the regulation of systemic lactate clearance, one of which may be linked to *Ucp1* down regulation. These data have been included as **new Figures 2C and D** and in the Results section (**please see page 11 of the revised manuscript with track changes**).

MINOR COMMENTS

1) Title - It would be important to add 'in male mice' to the title to make it clear to the reader the model studied please.

Reply. We thank the Reviewer with this suggestion and changed the title accordingly: “Brown adipose tissue activity impacts systemic lactate clearance in male mice”. We also specified that in the abstract.

2) In figure 4D, it would be helpful for the reader if the authors could write MCT1WT and MCT1ΔAd on the respective images please.

Reply. Changes have been done accordingly.

Referee #2:

In this study, Montané et al. investigated the impact of BAT stimulation on lactate uptake and intra-tissue metabolism. Male C57Bl/6J mice were housed at 21{degree sign}C, exposed to 4{degree sign}C during 7 days or housed at 30{degree sign}C for 2 months, then underwent i.p. lactate tolerance tests, without or with 13C-labeling, under fasted (6h) or fed conditions to perform metabolic isotope tracing experiments. The authors report that animals housed at 30{degree sign}C, whereby BAT is considered largely inactive and show reduced *Ucp1* expression, present with reductions in lactate clearance to a similar extent as *Ucp1ko* mice, while those housed at 4°C showed improved clearance compared to mice housed at 21°C. Housing temperature appeared to impact the gene expression of certain genes associated with lactate metabolism or proton-linked transport of metabolites like lactate or pyruvate, although *Mct1ko* mice did not present with any changes in lactate clearance. Finally, based on the metabolic isotope tracing experiments, the authors suggest that the enrichment pattern whereby there is a presence of m+1 and m+2 lactate species implies a cycling between pyruvate and lactate, which appears to be greater in mice housed at 21°C and 4°C than when housed at 30°C.

Critique

This study provides important observational insights into the uptake and possible metabolic fate of lactate, a seemingly important metabolite and substrate for BAT thermogenesis. There is mounting evidence demonstrating the role of lactate as a signaling molecule as well as a critical metabolic substrate, but little is known about how it is used once taken up by thermogenic organs like BAT. In this regard, this manuscript provides an important starting point to begin exploring the latter. However, there are important methodological and interpretational limitations that should be considered and possibly addressed.

Reply. We thank the Reviewer for their comments.

(1) Both the unlabeled and labeled lactate were administered by i.p. - this likely has a significant impact in the metabolism of lactate compared to i.v. administration. What was the justification for this administration and how does its metabolism differ from i.v.? There are apparent differences in lactate appearance between i.p. vs. s.c. administration, which is also sex-dependent (PMID: 31934509), is this also the case for i.p. vs. i.v. administration?

Reply. We decided to use intraperitoneal administration to avoid as much as possible the stress linked to the injection. Indeed, the tail vein of C57Bl6 mice is hardly visible and intravenous administration in this site requires specific and stressful interventions and handling (warming/massaging the tail, contention). While intravenous injection can also be done in the retro-orbital vein, this has to be done under anesthesia which is known to perturb both lactate metabolism (Horn et Klein, 2010, PMID: 20933036) and BAT activity (Ohlson et al. 1994, PMID: 8042786, Ohlson et al. 2003, PMID: 12552204). For all these reasons, we choose intraperitoneal injection which can be done in alert animals and which can be quickly completed, limiting the stress for the animals. Also, we opted for intraperitoneal versus subcutaneous injection, to be closer to the experimental conditions generally used for other tolerance tests such as glucose tolerance tests. We now better justify the choice of this mode of injection in the Material and Method section, referring to the above-mentioned articles (**please see page 7 of the revised manuscript with track changes**). In the Material and Methods section, we also pointed the fact that the mode of injection can have consequences on lactate appearance, citing the study mentioned by the Reviewer (Haugen et al. 2020, PMID: 31934509) and that interpretation of the results has to be done in the context of these intraperitoneal injections (**please see page 7 of the revised manuscript with track changes**). We also changed the title and abstract to better reflect that this study was only performed in male mice to ensure unambiguous sex reporting.

(2) The interpretation of the metabolic isotope tracing experiments are more complex than how it is presented. First, currently, there is a lack of evidence supporting the claims and an absence of confirmatory experiments to validate them. As it stands, it remains largely observational and open to many interpretations. The authors could help with interpretation by inhibiting pathways or silencing certain enzymes or transporters. Aside from using Mct1 knockout mice, this is the extent of probing that was done and these mice likely display some compensatory mechanisms.

Reply. We totally agree with the Reviewer that, at this point, our work provides observational findings regarding lactate utilization in different organs, and that metabolic isotope tracing alone cannot provide definitive mechanistic insights. Our first mechanistic hypothesis regarding the possible mechanism(s) behind the regulation of systemic lactate clearance was the involvement of the MCT1 transporter - strongly expressed by brown and beige adipocytes. We thus generated this new transgenic mice model (*Mct1*^{fllox} mice crossed with *adiponectin* CRE^{ERT2} mice) to test this hypothesis. Data showed no impact of MCT1 deletion in adipose tissues on systemic lactate clearance, maybe indeed because of compensatory mechanisms in this transgenic mice model as mentioned in the Discussion section and by the Reviewer. In response to the Reviewer's comment but also to that of Reviewer #1, we measured

mRNA levels of other *Mct* isoforms and found that they did not change (please see new **Figures 4D, 4E, 4F, 4G** of the manuscript and the Results section (**page 12 of the revised manuscript with track changes**)). However, this does not exclude the possibility that these transporters or other transporters (from the MCT family or from others families) present in adipose tissues could compensate for the loss of MCT1. While taking a long time to set up, we chose a genetic approach and not a pharmacological one because with the latter, tissue specificity cannot be controlled, rendering the interpretation of the data very complex. New transgenic mice models targeting other molecules (enzymes, transporters) in specific tissues should now be generated to better decipher the mechanism(s) regulating systemic lactate clearance. Given the time needed to generate new transgenic mice models, this work will be the purpose of a complete new and dedicated study and will be the object of a future work. We included in the Discussion and in the limits of the study the fact that additional work is needed to better decipher and validate the underlying mechanisms, notably through the generation of tissue-specific transgenic mice models (**please see pages 17/18 of the revised manuscript with track changes**).

Of note, isotope tracing data in liver and kidney that we added in response to Reviewer #1's comment (please see new **Figures 5A, 5B, 5E, 5F, 5G, 5H**, Results section (**pages 13/14 of the revised manuscript with track changes**)) and Discussion (**page 16 of the revised manuscript with track changes**)) reveal that lactate contribution to gluconeogenesis was increased in mice adapted to prolonged cold exposure - in accordance with the increased glucose generation from lactate upon acute cold exposure as recently demonstrated (Bornstein et al., 2023, PMID: 37802078), opening up hypotheses regarding mechanisms possibly involved in lactate clearance.

Second, there are critical products of lactate or pyruvate metabolism that are missing, such as acetyl CoA or citrate, carbon dioxide, malate, fumarate, aspartate (though it is in the supplemental files) and PEP. If referring to pyruvate carboxylation, then minimally, oxaloacetate should be presented as should malate and PEP.

Reply. We included in the initial version of the manuscript ¹³C enrichment data of several metabolites related to the TCA cycle including succinate, glutamine, glutamate, cis-aconitate, alpha-ketoglutarate as well as aspartate. Unfortunately, the metabolites cited by the Reviewer could not be detected in BAT. However, to provide a more complete view of metabolic pathways, we analyzed ¹³C enrichment in additional metabolites that could be detected in BAT such as glycerol-3-P, alanine, aconitate, 2 hydroxyglutarate, propionate and acetoacetate. While no modification of ¹³C labeling enrichment in acetoacetate, trans-aconitate and 2 hydroxyglutarate was observed between the different housing temperatures, a decreased ¹³C enrichment in glycerol-3-phosphate, alanine and propionate in BAT of 30°C mice compared to 4°C mice was observed (**please see Figure 1 for Reviewers**). These data further highlight that thermoneutral housing reduced lactate metabolism into diverse and specific metabolic pathways in BAT.

Figure 1 for reviewers

Third, gene expression tells us very little about the metabolism of an organ. There are many regulatory steps between gene expression and protein/enzymatic function. Minimally, we need to see protein levels and functional outcomes.

Reply. We fully agree with the Reviewer that gene expression alone provides limited insight into the metabolic activity of a tissue, due to the multiple steps of regulation between mRNA levels and protein content. Furthermore, protein level - and even transporter/enzymatic activity measured *ex vivo* - do not necessarily reflect actual metabolic activity *in vivo* and functional outcomes, due to complex post-translational regulation and context-dependent system dynamics (substrate availability, thermodynamic constraints, etc.). This is precisely why, in addition to transcriptomic analyses, we implemented isotopic tracing to directly monitor *in vivo* metabolic pathway activity and flux, which provides a functional readout of metabolism in real time. While deeper proteomic or enzymatic profiling could offer complementary insights, the number of potential targets and regulatory nodes involved makes this an extensive undertaking. Investigating the specific role and contribution of individual enzymes would ultimately require targeted intervention strategies (i.e., genetic inactivation), which are beyond the scope of the current study. We have now indicated this limitation and our plans for future mechanistic studies in the revised Discussion section and in the limits of the study (**please see pages 17/18 of the revised manuscript with track changes**).

(3) Reporting of data

- Given that ¹³C-lactate will be rapidly metabolized, the data reported and normalized to tissue ¹³C-lactate will be highly influenced by a denominator effect, making it very difficult to interpret. Why not present the fractional enrichments (as is done in Table S1), or normalize to the injected dose or relative to total ¹³C?

Reply. As tissues can use both labeled and unlabeled (endogenous) lactate, the contribution of the labeled lactate to the production of different metabolites requires the normalization of the ¹³C enrichment of the different metabolites to the ¹³C enrichment of the source. Also, as ¹³C lactate enrichment can vary between tissues and conditions, any labeling in downstream metabolites may be interpreted accordingly and it is therefore essential to normalize to the source in order to evaluate the metabolic fate of lactate. This normalization approach is commonly used in *in vivo* isotopic tracing studies, as shown in several recent publications (Wang et al. 2021 (PMID: 31918920), Wang et al. 2020 (PMID: 33043581), Cai et al. 2025, PMID: 39960461, Jimenez-Blasco et al. 2024 (PMID: 38789798) and can be applied to either specific isotopologues or to molecular ¹³C enrichments. As mentioned by the Reviewer, we also provided the raw fractional enrichment data for all metabolites (Table S1). We justify this better in the Results section (**please see page 13 of the revised manuscript with track changes**).

- There is an insufficient sample size to complete the statistics presented.

Reply. We thank the Reviewer for raising this important point. We fully agree that statistical power is a critical aspect of data interpretation. Our sample sizes were chosen based on standards commonly used in similar *in vivo* metabolic studies. However, in response to the Reviewer's comment and to strengthen our statistical analyzes, we re-done them, using procedures aligned with those applied in comparable studies (Cai et al. 2025, PMID: 39960461). As required by the journal guidelines, we now report the exact p values on the graphs. We also detailed the statistical test used for each panel in the corresponding Figure legends. This new statistical study does not drastically change the significance between the groups, and interpretations and conclusions remain the same as those previously established. Following is the detailed procedure, that is presented in the Material and Methods section (**please see page 10 of the revised manuscript with track changes**). Data were tested for normal distribution using the Shapiro-Wilk test. When distribution was normal, a parametric test was used. When data were not showing a normal distribution, data were log-transformed and tested for normality. If the transformed data showed a normal distribution, we performed parametric tests on the transformed data. If transformed data did not show normal distribution, we performed nonparametric tests on the nontransformed data. Comparison between three groups was done using one-way ANOVA with Welch correction if variances were not equal between groups, and with Tukey's or Dunnett's T3 multiple comparisons tests when variances were equal or not equal respectively. When data were not showing a

normal distribution, a non-parametric Kruskal-Wallis test with Dunn's multiple comparison test was performed for comparison between three groups. For lactate tolerance tests, a two-way ANOVA with Tukey's multiple comparisons test was performed. Comparison between two groups was performed with a Student's t-test with Welch's correction if variances were not equal between groups or with a Mann-Whitney test. N values are specified in the Figure legends. Exact p values are reported on the Figure legends. Differences were considered statistically significant at $p \leq 0.05$.

- Given the dynamics of ^{13}C -lactate, it would be important to look at both the mitochondrial and cytosolic pools of metabolites, not just the tissue as a whole, to make sense of what might be happening.

Reply. We agree with the Reviewer that this information would be very interesting. However, metabolism is so labile that high caution is made to stop the metabolic reactions very rapidly. Any experiments tempting to isolate mitochondria from the cytosol compartment that require different centrifugation steps would definitely impact ^{13}C labeling patterns.

- Standard deviation should be reported rather than SEM.

Reply. We replaced SEM by standard deviation (SD) in all the figures (and notified this in the Material and Methods section, **please see page 10 of the revised manuscript with track changes and in figure legends**).

(4) Methods

- As currently written, the Methods lack some important details - how were animals euthanized? Was lactate tolerance performed under anaesthesia or awake? If so, what anaesthetic was used, how long, what dose? Presumably, the lactate tolerance tests were performed at ambient temperature, rather than their housed temperatures? How could this impact the outcomes?

Reply. For all experiments except ^{13}C labeling tracing experiments, mice have been euthanized by cervical dislocation. For ^{13}C labeling experiments in which high amount of blood was needed, decapitation was required and performed, in accordance with ethical recommendations and after approval by the ethic committee.

Lactate tolerance tests were performed without anesthesia, at 21°C for all the three experimental groups. We chose this experimental condition to avoid the stress that could have been generated by moving cages every five minutes from their housing places to the experimental zone. One can reasonably postulate that differences could have been even more marked between the groups if the tolerance tests had been carried out under the respective housing temperatures conditions. We added precisions regarding these experimental conditions in the Material and Methods (**please see page 6 of the revised manuscript with track changes, new paragraph entitled "Ethic approval" and page 7**).

- Are the *Ucp1* ko mice on the C57 or 129/sv or hybrid background? The original *Ucp1* ko mice have serious mitochondrial defects (see PMID: 28630339). Although cited and referenced in the Discussion, it is unclear which model is ultimately used and why it would be selected over more appropriate *Ucp1* ko models currently available.

Reply. The *Ucp1*^{KO} mice used in our study are on a C57BL/6J background as originally generated by the group of Leslie Kozak (Enerbäck et al. 1997, PMID: 9139827). We acknowledge, however, the findings from Kazak et al. (Kazak et al. 2017, PMID: 28630339) regarding mitochondrial abnormalities in the original *Ucp1*^{KO} line. Our choice to use the C57BL/6J-*Ucp1*^{KO} model was based on its extensive characterization in previous thermogenesis and metabolic studies. Other *Ucp1*^{KO} mice exist, whatever in the 129S1/SvImJ and in the mixed 129S1/SvImJ × C57BL/6J backgrounds, this latter model exhibiting different cold sensitivity than the congenic models (Hofmann et al. 2001, PMID: 11279075). Interestingly, this model exhibits similar downregulation of mitochondrial components as the *Ucp1*^{KO}

mice in the C57BL6 background, as recently demonstrated (Rahbani et al. 2024, PMID: 38272036). While we recognize that newer inducible models may offer advantages in studying acute responses or avoiding potential compensation and/or secondary dysregulation, the C57BL/6J line with the constitutive UCP1 depletion allowed us to compare our findings with a broad existing literature and ensure a complete UCP1 depletion, without the possibility of incomplete, variable or time sensitive depletion that can sometimes occur with inducible systems. However, we agree that future studies using alternative *Ucp1*^{KO} models (on other genetic backgrounds or inducible models as recently published by Rahbani et al. 2024 (PMID: 38272036)) would be valuable to expand upon our findings. We added a comment on this in the Discussion section (**please see page 15 of the revised manuscript with track changes**).

- Pg. 12, 1st paragraph - the authors indicate that "Because 13C lactate was the only 13C source, the relative contribution from the tracer to downstream metabolites was calculated by dividing the enrichment of the metabolite with the enrichment of the tracer (Wang et al., 2020b)." Is this in reference to enrichment of tracer in plasma, in tissue or of infusate?

Reply. Normalized molecular ¹³C-enrichments in metabolites within the different tissues or within the plasma were calculated by normalizing to the molecular ¹³C-enrichment of lactate within the different tissues or within the plasma respectively, for the same animal. We have made this clearer in the Material and methods section and in the Figure legends (**please see page 9 of the revised manuscript with track changes**).

- If animals were group housed, ambient temperature (21C) may not be that cold.

Reply. Indeed, the temperature inside the cages can be warmer than in the housing room. In the recent article published by Rahbani et al. (PMID: 38272036), the temperature inside the cages of mice housed in standard conditions (23°C ± 1°C) has been reported to be ~2°C higher than the temperature set in the room. However, this remains far from the thermoneutral state for mice, consistently with the intermediate activation of brown and beige adipose tissues of 21°C mice compared to 4°C and 30°C mice (Figure 1).

- It is still unclear how pyruvate cycling is calculated? The cycle and how it is determined needs to be described more completely. Simply an (m+1 + m+2)/m+3 calculation is not a reflection of pyruvate cycling.

Reply. We understand from the Reviewer's comment that the pyruvate cycling was not well described enough and that the term "pyruvate cycling index" applied to the m+1+m+2/m+3 ratio was somehow misleading. We made substantial modifications to improve these aspects.

To make interpretation clearer, we included data regarding pyruvate isotopologues distribution across tissues and conditions that show similar pattern than for lactate (**please see new Figure 6B and Results section page 14 of the revised manuscript with track changes**). The pyruvate cycle has been previously documented (Hasenour et al. 2020, PMID: 32755580), and indicates that the m+1 and m+2 pyruvate molecules are necessarily derived from pyruvate recycling, and results also in lactate recycling when the recycled pyruvate is converted back into lactate. Under isotopic steady-state conditions, assuming that the m+3 lactate molecules are derived only from the administrated lactate, the (m+1+m+2)/m+3 ratio can be used to calculate the actual extent of pyruvate recycling. But in our work, we are not in such conditions, so this ratio is only a proxy of pyruvate recycling (the higher the ratio, the higher pyruvate is recycled), not a true calculation of it. Because the term "pyruvate recycling index" was misleading regarding that point, in the revised version we have removed this term in the figures and in the text (and replaced it by (m+1+m+2)/m+3 labeling ratio), and made it clearer in the text that the ratio was a proxy of pyruvate recycling. We also now tried to better explain the metabolic pathways associated with the pyruvate cycling. Below is the complete description of the pyruvate cycling that has

been included in the **Results** section (please see pages 14/15 of the revised manuscript with track changes); please note that the new **Figure 6** has been adapted accordingly.

*“The occurrence of $m+1$ and $m+2$ species is due to pyruvate cycling, which has been described in other tissues such as the brain (Cerdan, 2017), and is a metabolic process where pyruvate is used and re-synthesized. It occurs via mechanisms (**Figure 6C**) that include i) the carboxylation of pyruvate into oxaloacetate via the pyruvate carboxylase (causing the introduction of ^{12}C atoms into the initially triply labeled pyruvate, **Figure 6D, left panel**), ii) full oxaloacetate rotation into the TCA cycle (causing further isotope dilution and loss of label after decarboxylation and label scrambling in oxaloacetate or malate) or the reverse equilibrium of oxaloacetate with other four-carbon intermediates of the TCA cycle (causing label scrambling in oxaloacetate or malate) (**Figure 6D, left panel**) and iii) the conversion of either oxaloacetate (via PEPCK + pyruvate kinase) or malate (via the malic enzyme) by decarboxylation back into pyruvate (**Figure 6C, 6D**). As a result, the recycled pyruvate can have lost one ($m+2$ species) or two ($m+1$ species) ^{13}C atoms (**Figure 6D, left panel**) as previously demonstrated (Hasenour et al., 2020). The $m+1$ and $m+2$ isotopologues can be also obtained if pyruvate enters the TCA cycle via the pyruvate dehydrogenase activity (**Figure 6D, right panel**), and is further re-formed from cytosolic malate or oxaloacetate, a process that requires the additional activity of an anaplerotic activity to replenish the TCA cycle, which is mostly pyruvate carboxylase in mammalian cells. In both cases, there is utilization and re-synthesis of pyruvate, which can be evaluated from the $(m+1+m+2)/m+3$ labeling ratio, considered as a proxy of pyruvate cycling (the higher the ratio, the higher the recycling). Although no modification occurred in 4°C mice compared to 21°C mice, we observed a strong decrease in the pyruvate $(m+1+m+2)/m+3$ labeling ratio in iBAT at thermoneutrality and no changes in plasma, SCAT, liver and kidney (**Figure 6E**). Similar profiles were found for the lactate $(m+1+m+2)/m+3$ labeling ratio indicating that lactate cycling occurs as well (**Figure 6F**).”*

Specific comments:

(1) Pg 4, ln 1 - thermogenesis and heat production are synonymous, so thermogenesis can't trigger heat production.

Reply. Change has been done accordingly (please see page 4 of the revised manuscript with track changes and below).

“In cold environments, mammals maintain constant body temperature through thermogenesis, a metabolic process that produces heat (Lowell & Spiegelman, 2000; Ricquier, 2006).”

(2) Pg 4, 1st paragraph - BAT doesn't become inactive when heat production and heat loss are balanced. It becomes inactive when the sympathetic stimulation is reduced, which would happen when stimulus (heat loss) is removed.

Reply. Change has been done accordingly (please see page 4 of the revised manuscript with track changes and below).

Introduction (page 4):

“In rodents, non-shivering thermogenesis mainly occurs in brown adipose tissue (BAT), that dissipates energy as heat, and becomes inactivated when the sympathetic stimulation is reduced, which would happen when stimulus (heat loss) is removed, in a state referred to as thermoneutrality.”

(3) Pg 4, 2nd paragraph - UCP1 uncouples cellular respiration from ATP synthesis.

Reply. Change has been done accordingly (please see page 4 of the revised manuscript with track changes and below).

Introduction (page 4):

“BAT thermogenesis is driven by the high mitochondrial content of brown adipocytes and by the expression of the mitochondrial uncoupling protein-1 (UCP1) (Nicholls, 1976; Ricquier & Kader, 1976), which uncouples cellular respiration from ATP synthesis once activated notably by long chain fatty acids (Nicholls & Locke, 1984).”

(4) Pg 4, 2nd paragraph - beige adipocytes don't produce the same amount of heat as brown adipocytes and seem to also rely on other futile cycles like calcium cycling and creatine cycling.

Reply. We included this notion and added a recent article (Bunk et al. 2025, PMID: 40185737) showing that futile cycles also exist in brown adipocytes (**please see page 4 of the revised manuscript with track changes and below**).

Introduction (page 4):

“While UCP1 expressed by beige adipocytes is thermogenically competent, the thermogenic capacity of beige depots remains lower than those of the brown depot at the whole systemic level (Shabalina et al., 2013). Recently, several UCP1-independent mechanisms, including cycling of creatine, lipid and calcium, were reported to support the energy-dissipating properties of brown and beige adipocytes (Mottillo et al., 2014; Kazak et al., 2015; Ikeda et al., 2017; Roesler & Kazak, 2020; Oeckl et al., 2022; Sharma et al., 2024; Bunk et al., 2025), highlighting the complexity and versatility of metabolic pathways involved in the function of thermogenic adipose tissues.”

(5) Pg. 10 - Ldha and Ldhb mRNA are not enzymes and therefore cannot be involved in the reversible conversion of pyruvate into lactate.

Reply. We made the change accordingly (**please see page 12 of the revised manuscript with track changes and below**).

Results section (page 12):

“We found that mRNA levels of Ldha and Ldhb – which encode enzymes involved in the reversible conversion of pyruvate into lactate - as well as those of the lactate transporter Mct1 - were very sensitive to thermogenic conditions”.

(6) Pg 12 - 2nd paragraph - authors state that "Because ^{13}C lactate was the only ^{13}C source, the relative contribution from the tracer to downstream metabolites was calculated by dividing the enrichment of the metabolite with the enrichment of the tracer (Wang et al., 2020b)." However, lactate is rapidly metabolized, so why would we assume that 15 min post injection under possibly stimulated states, we would mainly expect lactate isotopologue to be m+3 lactate?

Reply. We would like to thank the Reviewer for this comment. Our initial sentence *“Because injected lactate was uniformly labeled on its 3 carbons, the main expected lactate isotopologue was m+3 lactate”* was confusing and we removed it. We were actually expecting similar lactate isotopologues distribution across tissues. To better explain this, we included in the new version of the manuscript lactate (and pyruvate) isotopologues distribution in plasma but also in liver and kidney.

The labeling experiments were set-up to help explaining the metabolic mechanisms involved in lactate clearance during the lactate tolerance test. The 15 min time-point was selected because it corresponds to a time for which lactate is decreasing during the test, and is relevant to examine the metabolic processes underlying this clearance. Hence, and as stated by the Reviewer, lactate is rapidly metabolized at this time-point. The major labeled isotopologue to be detected in plasma lactate was the m+3 in all three conditions (**Figure 6A**). This isotopologue is likely coming from the administered $[\text{U-}^{13}\text{C}]$ -lactate. However, we also observed m+1 (singly labeled) and m+2 (doubly labeled) lactate species specially in iBAT where they were present at level similar to the triply labeled one in the 4°C group. The lactate m+1 and m+2 isotopologues could be also detected in liver and kidney, though at a lower extent than in iBAT, but were almost undetected in SCAT. Similar isotopologue profiles were observed for pyruvate (**Figure 6B**). Changes have been done in the Results section (**please see page 14 of the revised**

manuscript with track changes) and the **new Figure 6** has been completed with data of plasma, liver and kidney.

(7) Pg 12 - 2nd paragraph - couldn't a singly and doubly labeled species reflect a loss of carbon to CO₂?

Reply. If decarboxylation occurs, this generates a new molecule with less atoms of carbons. However, we identified single and doubly labeled species within the lactate (and pyruvate) pool, reflecting utilization and re-synthesis of these molecules, and not a direct carbon loss of these molecules. We made this clearer in the Results section (**please see page 14 of the revised manuscript with track changes and below**).

Results section (page 14):

“However, we also observed $m+1$ (singly labeled) and $m+2$ (doubly labeled) lactate species, - i.e. one and two ¹³C carbons among the three carbon atoms of lactate species - especially in iBAT.”

Dear Dr Carriere,

Re: JP-RP-2025-288871R1 "**Brown adipose tissue activity impacts systemic lactate clearance in male mice**" by Rémi Montané, Yannick Jeanson, Damien Lagarde, Spiro KHOURY, Léana Porcher-Bibes, Jean Nakhle, Marie Sallèse, Mélissa Parny, Isabelle Raymond-Lebron, Emma Huard, Raphael Alves de Souza, Anne Galinier, Luc Pellerin, Anne-Karine Bouziers Sore, Jean-Philippe Pradère, Cedric Moro, Louis Casteilla, Armelle Yart, Cedric Dray, Jean-Charles Portais, Isabelle Ader, and Audrey Carriere

Thank you for submitting your manuscript to The Journal of Physiology. It has been assessed by a Reviewing Editor and by 2 expert referees and we are pleased to tell you that it is acceptable for publication following satisfactory revision.

REVISION CHECKLIST:

Please upload two versions of your manuscript text: one with all relevant changes highlighted and one clean version with no changes tracked. The manuscript file should include all tables and figure legends, but each figure/graph should be uploaded

as separate, high-resolution files. The journal is now integrated with Wiley's Image Checking service. For further details, see: <https://www.wiley.com/en-us/network/publishing/research-publishing/trending-stories/upholding-image-integrity-wileys-image-screening-service>

We look forward to receiving your revised submission.

Yours sincerely,

Paul Greenhaff
Senior Editor
The Journal of Physiology

EDITOR COMMENTS

Reviewing Editor:

Thank you for your thoughtful responses to the referee comments. You have satisfactorily addressed their concerns.

Senior Editor:

Thank you for the revised manuscript. The Reviewing Editor and Reviewers are of the opinion that the authors have done a very good job at revising the manuscript and believe that the paper will be quite influential.

One small (but important) further modification is required. Please can the authors provide evidence in the section "Ethic approval" that decapitation without anaesthesia was the most appropriate method of killing in the study? The animal ethical principles under which the Journal of Physiology operates states decapitation without anesthesia is only appropriate if no other method is appropriate. The Senior Editor is not questioning that it isn't appropriate, they would just like evidence included to verify that this was indeed the case in this particular study. Thank you.

REFeree COMMENTS

Referee #1:

I thank the authors for addressing all of my comments and for the additional work that has improved the manuscript. No further comments.

Referee #2:

I commend the authors for their thorough responses to the first rounds of questions and consider it acceptable for publication in the Journal of Physiology

END OF COMMENTS

EDITOR COMMENTS

Reviewing Editor:

Thank you for your thoughtful responses to the referee comments. You have satisfactorily addressed their concerns.

Senior Editor:

Thank you for the revised manuscript. The Reviewing Editor and Reviewers are of the opinion that the authors have done a very good job at revising the manuscript and believe that the paper will be quite influential.

Reply. We thank the Reviewing editor and the Senior editor for their comments.

One small (but important) further modification is required. Please can the authors provide evidence in the section "Ethic approval" that decapitation without anaesthesia was the most appropriate method of killing in the study? The animal ethical principles under which the Journal of Physiology operates states decapitation without anesthesia is only appropriate if no other method is appropriate. The Senior Editor is not questioning that it isn't appropriate, they would just like evidence included to verify that this was indeed the case in this particular study. Thank you.

Reply. As requested, we have modified the Ethic approval section to better justify the method of euthanasia (please see page 6 of the revised manuscript).

“For ¹³C labeling experiments, decapitation was performed without anesthesia in accordance with ethical recommendations and after approval by the ethic committee. This method was the most appropriate because it enabled to quickly harvest a large quantity of blood without anesthesia, avoiding the likely interference of anesthetic drugs on tissue and blood metabolic profiles (Pierozan et al., 2017).”

REFeree COMMENTS

Referee #1:

I thank the authors for addressing all of my comments and for the additional work that has improved the manuscript. No further comments.

Referee #2:

I commend the authors for their thorough responses to the first rounds of questions and consider it acceptable for publication in the Journal of Physiology.

Reply. We thank the Referees for their comments.

Dear Dr Carriere,

Re: JP-RP-2025-288871R2 "**Brown adipose tissue activity impacts systemic lactate clearance in male mice**" by Rémi Montané, Yannick Jeanson, Damien Lagarde, Spiro KHOURY, Léana Porcher-Bibes, Jean Nakhle, Marie Sallèse, Mélissa Parny, Isabelle Raymond-Letron, Emma Huard, Raphael Alves de Souza, Anne Galinier, Luc Pellerin, Anne-Karine Bouziers Sore, Jean-Philippe Pradère, Cedric Moro, Louis Casteilla, Armelle Yart, Cedric Dray, Jean-Charles Portais, Isabelle Ader, and Audrey Carriere

We are pleased to tell you that your paper has been accepted for publication in The Journal of Physiology.

Yours sincerely,

Paul Greenhaff
Senior Editor
The Journal of Physiology

If you would like to receive our 'Research Roundup', a monthly newsletter highlighting the cutting-edge research published in The Physiological Society's family of journals (The Journal of Physiology, Experimental Physiology, Physiological Reports, The Journal of Nutritional Physiology and The Journal of Precision Medicine: Health and Disease), please click this link, fill in your name and email address and select 'Research Roundup':
<https://www.physoc.org/journals-and-media/membernews>

- You can help your research get the attention it deserves! Check out Wiley's free Promotion Guide for best-practice recommendations for promoting your work at: www.wileyauthors.com/eeo/guide. You can learn more about Wiley Editing Services which offers professional video, design, and writing services to create shareable video abstracts, infographics, conference posters, lay summaries, and research news stories for your research at: www.wileyauthors.com/eeo/promotion.

EDITOR COMMENTS

Senior Editor:

Thank you for making the final amendment to the manuscript. Congratulations and thank you for submitting your work to The Journal of Physiology.